THE
EMBO
JOURNAL

# Unloading of homologous recombination factors is required for restoring double-stranded DNA at damage repair loci

Yulia Vasianovich[1,†], Veronika Altmannova[2,3], Oleksii Kotenko[1], Matthew D Newton[1], Lumir Krejci[2,3,4] & Svetlana Makovets[1,*]

## Abstract

Cells use homology-dependent DNA repair to mend chromosome breaks and restore broken replication forks, thereby ensuring genome stability and cell survival. DNA break repair via homology-based mechanisms involves nuclease-dependent DNA end resection, which generates long tracts of single-stranded DNA required for checkpoint activation and loading of homologous recombination proteins Rad52/51/55/57. While recruitment of the homologous recombination machinery is well characterized, it is not known how its presence at repair loci is coordinated with downstream re-synthesis of resected DNA. We show that Rad51 inhibits recruitment of proliferating cell nuclear antigen (PCNA), the platform for assembly of the DNA replication machinery, and that unloading of Rad51 by Srs2 helicase is required for efficient PCNA loading and restoration of resected DNA. As a result, srs2Δ mutants are deficient in DNA repair correlating with extensive DNA processing, but this defect in srs2Δ mutants can be suppressed by inactivation of the resection nuclease Exo1. We propose a model in which during re-synthesis of resected DNA, the replication machinery must catch up with the preceding processing nucleases, in order to close the single-stranded gap and terminate further resection.

**Keywords** DNA re-synthesis; PCNA; Rad51; recombination machinery; Srs2
**Subject Categories** DNA Replication, Repair & Recombination
**The EMBO Journal (2017) 36: 213–231**

## Introduction

In both prokaryotes and eukaryotes, DNA double-stranded breaks (DSBs) predominantly occur as a result of broken replication forks (Vilenchik & Knudson, 2003). DSBs can also be generated due to DNA exposure to toxic chemicals or radiation as well as introduced by endogenous nucleases during developmentally programmed mechanisms such as meiosis and yeast mating type switching. DSBs are routinely repaired either by direct ligation of broken ends or by homology-dependent mechanisms such as homologous recombination (HR), break-induced replication (BIR) and single-strand annealing (SSA) (Symington *et al*, 2014). Alternatively, telomerase, the enzyme responsible for telomere maintenance (Greider & Blackburn, 1987), can interfere with repair by adding telomeric repeats to a DSB in a process called *de novo* telomere addition (Schulz & Zakian, 1994). Failure to repair DSBs results in decreased cell viability, particularly after exposure to DNA-damaging agents, increased gross chromosomal rearrangements and cancer predisposition underlying the biological significance of DNA repair mechanisms.

Homology-dependent DSB repair is highly conserved in eukaryotes. In yeast *Saccharomyces cerevisiae,* it involves (i) initial DSB processing by MRX(Mre11-Rad50-Xrs2)/Sae2 producing a short 3′ overhang; (ii) long-range DNA resection by two redundant machineries, Dna2/Sgs1-Top3-Rmi1 and Exo1 nuclease (Mimitou & Symington, 2008; Zhu *et al*, 2008), which generate long tracts of ssDNA covered by the ssDNA-binding protein RPA and required for DNA damage checkpoint activation and loading of homologous recombination machinery (Zou & Elledge, 2003; Lisby *et al*, 2004); (iii) loading of the homologous recombination protein Rad52 followed by recruitment of Rad51 which generates a nucleoprotein filament stabilized by Rad55/57 (Symington *et al*, 2014). During HR and BIR, Rad52/51/55/57 promote homology search and invasion of intact donor dsDNA by the processed broken end to initiate repair (Anand *et al*, 2013; Symington *et al*, 2014). In contrast, SSA does not require DNA external to the broken chromosome as homologous sequences on either side of the break provide complementarity between the processed ends and Rad52, but not Rad51/55/57, catalyse the strand annealing (Fishman-Lobell *et al*, 1992; Ivanov *et al*, 1996).

1 Institute of Cell Biology, School of Biological Sciences, University of Edinburgh, Edinburgh, UK
2 Department of Biology, Masaryk University, Brno, Czech Republic
3 International Clinical Research Center, St. Anne's University Hospital in Brno, Brno, Czech Republic
4 National Centre for Biomolecular Research, Masaryk University, Brno, Czech Republic
*Corresponding author. Tel: +44 131 650 5333; E-mail: smakovet@staffmail.ed.ac.uk
†Present address: Department of Microbiology and Infectious Diseases, Université de Sherbrooke, Sherbrooke, QC, Canada

However, HR can be also toxic emphasizing the need for its tight regulation. The Srs2 helicase inhibits HR machinery by disassembling Rad51 filament and reducing DNA extension, as demonstrated *in vitro* (Burkovics *et al*, 2013; Krejci *et al*, 2003; Veaute *et al*, 2003). This function is believed to be important for repression of excessive recombination, particularly at replication forks where Srs2 is recruited and regulated through its C-terminal domain (Papouli *et al*, 2005; Pfander *et al*, 2005; Burgess *et al*, 2009). Loss of Srs2 results in a paradoxical phenotype. On one hand, *srs2* mutants are hyper-recombinogenic (Aguilera & Klein, 1988), and on the other hand, they are deficient in DSB repair via HR and SSA (Vaze *et al*, 2002; Saponaro *et al*, 2010). Here we elucidate at the molecular level the role of Srs2 in multiple repair mechanisms involving extended DNA resection by showing that Srs2 is capable of dislodging Rad51 from ssDNA in order to promote loading of proliferating cell nuclear antigen (PCNA) and DNA replication machinery to restore dsDNA at repair loci. This function is distinct from the role of Srs2 at replication forks and essential for completion of DNA repair involving extended resection.

# Results

## Srs2 is not required for DNA damage checkpoint inactivation

Cell death of *srs2* mutants undergoing DSB repair is accompanied by accumulation of ssDNA and persistent activation of the DNA damage response (DDR) (Vaze *et al*, 2002; Yeung & Durocher, 2011). In order to distinguish between the defects of *srs2* mutants in DNA repair and the recovery from DDR, we designed a system in which DSB induction led to activation of DDR, but DNA repair was not required for cells to survive DSBs (Fig 1A). In this system, one side of the break contained 81 bp of $(TG_{1-3})_n$ telomeric sequence which protected the centromere-proximal DNA end from resection while the other side contained either 2 or 20 kb of non-essential DNA. Only 20 kb, but not 2 kb, should be long enough to generate sufficient ssDNA post-resection to activate DDR. When the 20-kb terminal fragment becomes completely degraded, the ssDNA as a signal for checkpoint activation disappears: if cells are

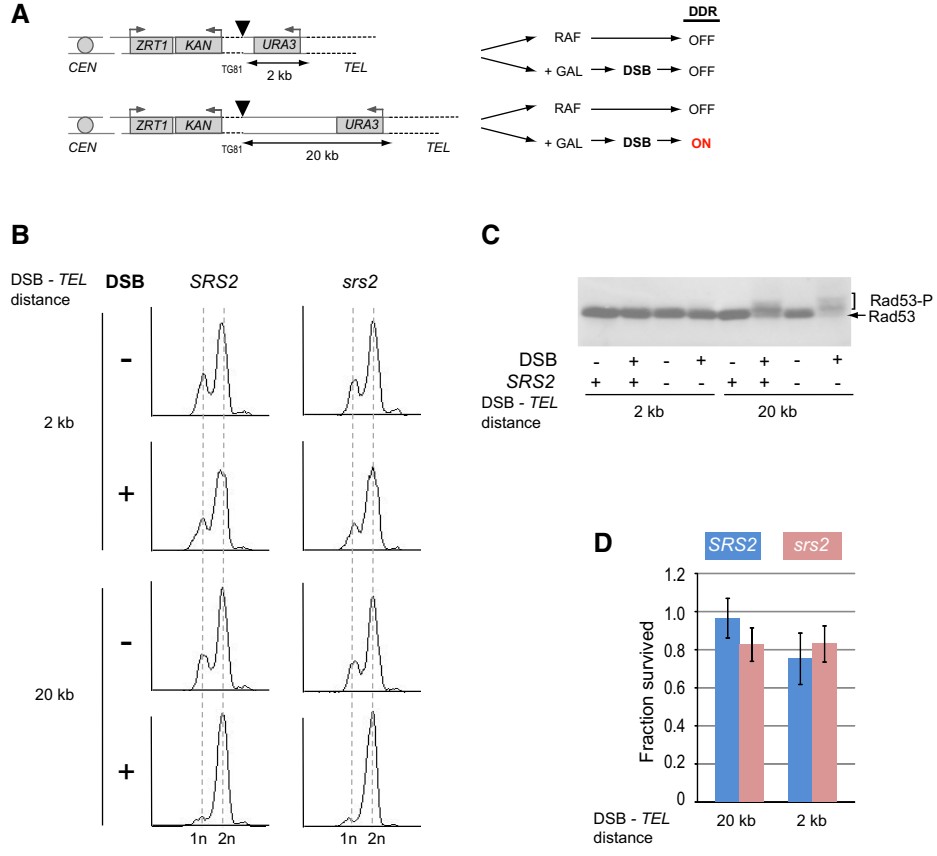

**Figure 1. Srs2 is not required for the recovery from the DNA damage-induced arrest.**

A   On the left, schematic of chr.VIIL variants, either with 2 or with 20 kb between a DSB and a telomere, used to study the effect of DNA damage checkpoint activation on cell viability in panels (B–D). Triangles represent HO sites, dashed lines represent telomeric sequences, TG81 represents 81 bp of $(TG_{1-3})_n$, and grey boxes represent genes with the grey arrows above showing promoters. The diagram on the right outlines the DNA damage response (DDR) activation as the reaction of the two different strains on DSB induction by the addition of galactose (GAL).

B   Analysis of cell cycle arrest (in G2) in response to DSB induction assayed by flow cytometry.

C   Rad53 phosphorylation (Rad53-P) in response to DSB assayed by Western blotting.

D   Cell survival upon DSB induction. Average ± SD (*n* = 4) is shown for each genotype.

Data information: Strains used: NK4230, NK4231; NK4264, NK4265; NK1949; NK4268, NK4269.

capable of checkpoint inactivation, they should be able to resume cycling.

Activation of DDR after DSB induction was assayed by Western blotting of Rad53, the key DNA damage signalling kinase, which becomes hyper-phosphorylated in response to DNA damage. We also used FACS analysis to ask whether cells accumulate in G2 as a result of DDR activation. As expected, DSB induction in both wild-type and srs2Δ strains resulted in activation of DDR in cells with 20 kb between the break and the telomere, but not when this distance was much shorter (Fig 1B and C). However, both *SRS2* and srs2Δ efficiently recovered from the cell cycle arrest as their survival was not affected by DSB induction (Fig 1D). Therefore, Srs2 is not required for the recovery from the DNA damage-induced arrest *per se* and the previously observed cell death of srs2Δ (Vaze *et al*, 2002) might come from the inability to complete DNA repair. Therefore, we next focused on the role of Srs2 in DSB repair by different mechanisms: we analysed *de novo* telomere addition, BIR and SSA in *SRS2* and srs2Δ cells.

## Analysis of *de novo* telomere addition in *SRS2* and *srs2* mutant cells

*De novo* telomere addition was assayed in *SRS2* and srs2Δ using a previously described genetic test (Makovets & Blackburn, 2009) involving a single galactose-inducible DSB (Fig 2A). Because *de novo* telomere addition normally occurs with a very low frequency due to telomerase inhibition by Pif1 (Schulz & Zakian, 1994), the *pif1-m2* background was used in the genetic assay. In srs2Δ, *de novo* telomere addition was reduced ~47-fold, but this effect was completely suppressed by additional deletions of *RAD52*, *RAD51*, *RAD55* or *RAD57* (Fig 2B). These data suggest that the presence of the HR machinery at DSBs may inhibit *de novo* telomere addition and that the Srs2-dependent removal of the HR proteins might reverse this inhibition.

*De novo* telomere addition involves (i) extension of the 3′-end as a result of addition of telomeric $TG_{1-3}$ repeats by telomerase and (ii) synthesis of the complementary strand (C-strand) by the conventional replication machinery. In order to find out whether Srs2 is required at the earlier or the later step of this process, we first compared the addition of the telomeric $TG_{1-3}$ repeats to the 3′-end of a break in *SRS2* and srs2Δ. Cells with a galactose-inducible DSB were grown in YP + raffinose to mid-log phase and upon addition of galactose to the medium cell aliquots were taken for DNA analysis. qPCR was used to monitor addition of telomeric repeats through the time-course. One of the primers in the reaction was telomere-specific, that is consisted of $AC_{1-3}$ repeats (Fig 2C), and therefore, the PCR product could be formed only after telomerase-dependent extension of the 3′-end of the break. The other primer annealed 168 bp away from the HO-cleavage site as most of the *de novo* telomeres in *pif1-m2* are added close to the breakpoint (Schulz & Zakian, 1994). Consistent with the previously established functions of telomerase and Pif1, no addition of $TG_{1-3}$ repeats to DSBs was detected in wild-type cells, where telomerase is inhibited by Pif1 (Fig 2D, dark blue), and telomerase-deficient *pif1-m2 est2Δ* control (Fig 2D, orange). In contrast, addition of the $TG_{1-3}$ repeats in the *pif1-m2* telomerase-positive yeast was readily observed (Fig 2D, light blue) and was not affected by the lack of either Srs2 (Fig 2D, pink) or Rad51/52 (Fig 2D, green). Therefore, Srs2 is not required for the telomerase-dependent addition of $TG_{1-3}$ repeats to DSBs.

For the completion of *de novo* telomere addition, the complementary C-strand needs to be synthesized all the way to the resected 5′-end. In order to monitor the conversion of the ssDNA into dsDNA, we used a previously reported approach based on digestion of qPCR template with restriction enzymes in order to differentiate between ssDNA and dsDNA (Zierhut & Diffley, 2008): if the template is single-stranded, that is synthesis of the complementary strand has not occurred, then it cannot be cleaved by a restriction enzyme. By comparing relative amounts of template DNA in parallel qPCRs with and without restriction digestion, fractions of ssDNA and dsDNA in the template DNA can be calculated as explained in Materials and Methods. Time-course experiments, where G1-arrested *SRS2* and srs2Δ cells were subjected to DSB induction 1 h prior to S/G2 release into YP + galactose with nocodazole, were used to monitor the progress of *de novo* telomere addition both at the stage of $TG_{1-3}$ repeat synthesis by telomerase and during conversion of ssDNA into dsDNA at the break. Consistent with the experiments in non-synchronized cells (Fig 2D), srs2Δ had no defect in addition of $TG_{1-3}$ repeats by telomerase: during the earlier time points, the repeat addition was even more efficient in the mutants than in *SRS2* (Fig 2E and F). However, when *Psi*I restriction enzyme was used to digest DNA templates prior to PCRs, a significant difference between *SRS2* and srs2Δ in the DNA status at the breaks healed by telomerase was observed. The mutant cells consistently had higher fractions of ssDNA at multiple time points (Fig 2E and F), suggesting that conversion of ssDNA into dsDNA during *de novo* telomere addition was delayed in srs2Δ mutants. Thus, Srs2 is required for the conversion of the ssDNA into dsDNA after telomerase-dependent addition of $TG_{1-3}$ repeats to the 3′-end and the reduced frequency of *de novo* telomere addition in srs2Δ in the genetic assay (Fig 2B) can be explained by the mutants' inability to restore dsDNA required in order to complete the repair.

## Srs2 is required for restoration of resected DNA during DSB repair by BIR

Repair of DSBs via BIR involves extensive DNA resection at the break locus in order to expose ssDNA regions which are essential for the search of intact homologous sequences. The efficiency of BIR among other factors depends on the extent of homology between broken DNA ends and donor chromosomes. In order to monitor BIR by Southern blotting, we constructed a system where the usage of BIR to repair a galactose-inducible DSB was very high due to the long (~6.3 kb) homology between the broken end on chr.VIIL and the homologous sequence on chr.II (Fig 3A). In a corresponding genetic assay, ~60% of wild-type cells survived DSB induction by using BIR for repair. BIR in isogenic srs2Δ mutants was reduced to ~30% (data shown below as part of Fig 7A).

In order to analyse progression of BIR in *SRS2* and srs2Δ, a DSB on chr.VIIL was induced by expression of the *HO* endonuclease gene from a *GAL* promoter in yeast cultures arrested in G1. One hour after the *HO* induction, cells were released from the arrest into YP + galactose with nocodazole to prevent cell cycle progression of cells with repaired breaks. Both re-synthesis of resected DNA and BIR-dependent duplication of the chr.II fragment downstream of the homology region were monitored by quantitative Southern blotting (Fig 3B–E, respectively). Break resection is expected to convert dsDNA into ssDNA which should lead to a decrease in the

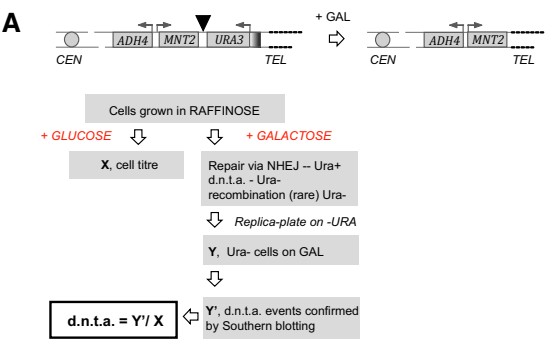

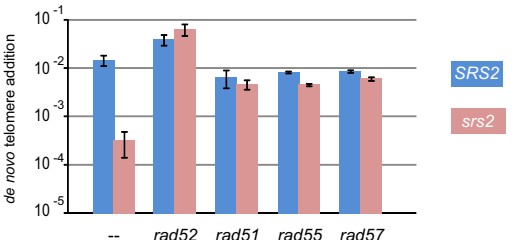

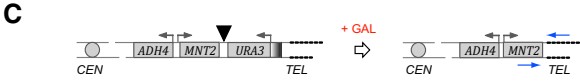

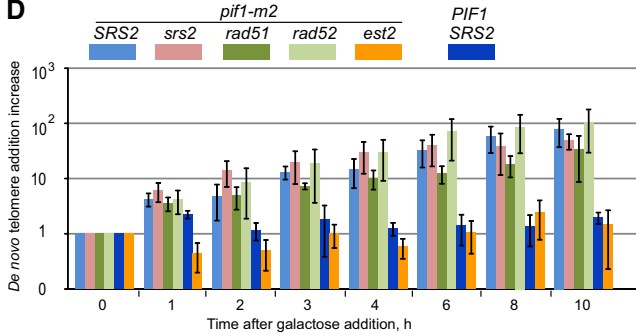

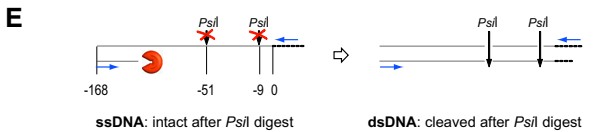

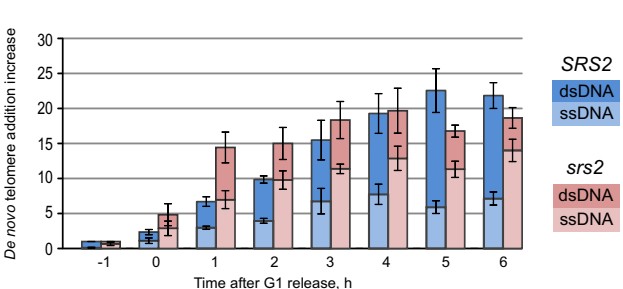

**Figure 2.  Srs2 is required to restore dsDNA during *de novo* telomere addition.**

A   Schematic of the genetic *de novo* telomere addition assay used in (B). Cells with a galactose-inducible HO-cut are grown on YP agar with raffinose prior to plating appropriate dilutions on YPD (no DSB induction) to score cell titre, and YP with galactose to induce HO expression and DSBs. DSB repair via *de novo* telomere addition leads to *URA3* loss and the *ADH4-MNT2* locus becoming part of terminal restriction fragments containing telomeres, which can be assayed by Southern blotting. Triangle represents the HO site.

B   Srs2 requirement in *de novo* telomere addition in cells with and without functional HR. All strains are *pif1-m2*. Strains used: NK1264, NK2375, NK2376; NK2014, NK2015; NK2451, NK2452, NK2012, NK2013; NK2457, NK2458; NK2363, NK2364; NK2469; NK2369, NK2370; NK2473–2475. Average ± SD (*n* = 3 or more) is shown for each genotype.

C   Schematic of the qPCR assay used in (D) to monitor (TG$_{1-3}$)$_n$ addition to DSBs. Triangle represents HO site, and blue arrows represent qPCR primers.

D   Dynamics of (TG$_{1-3}$)$_n$ addition monitored by qPCR through a time-course experiment (asynchronous populations). The *y*-axis shows a fold increase in *de novo* telomere-specific PCR product relative to the background levels at 0 h and normalized against an internal control (*ARO1*). Average ± SD (*n* = 3) is shown for each time point of each genotype. Strains used: NK3292, NK3293; NK4670, NK4671; NK4112, NK4113; NK4114, NK4115; NK3292 *est2Δ*, NK3293 *est2Δ*; NK4232, NK4233.

E   Schematic of the qPCR assay coupled with *Psi*I digestion used in (F) to quantify ssDNA/dsDNA ratio at the *de novo* telomere addition locus. Numbers indicate positions of *Psi*I restriction sites and qPCR primer sequences relative to DSBs. Blue arrows represent qPCR primers, and dashed lines represent telomeres.

F   Comparative analysis of ssDNA/dsDNA at *de novo* telomere addition loci in *SRS2* and *srs2Δ* during a time-course experiment (synchronized populations). The *y*-axis shows a fold increase in *de novo* telomere-specific PCR product relative to the background levels at 0 h and normalized against an internal control (*ARO1* locus). Average ± SD (*n* = 3) is shown for each time point of each genotype. Top set of error bars represents SD in relative increase of the *de novo* telomere-specific PCR product (as in panel D), while the lower set of error bars corresponds to quantifications of ss/dsDNA fractions. Strains used: NK3292, NK3293; NK4670, NK4671.

hybridization signal for the corresponding restriction fragment (as ssDNA is not cut by restriction enzymes), while re-synthesis of resected DNA should restore dsDNA and the hybridization signal at the analysed locus. Analysis of DNA dynamics at three different loci on chr.VIIL, 2.6, 6.8 and 15.2 kb away from the break, showed that *srs2Δ* mutants had a severe defect in restoration of resected DNA (Fig 3B and C). At the 2.6 and 6.8 kb loci, the fraction of cells with dsDNA status was much lower than in the *SRS2* population although the delayed restoration of dsDNA can be seen at 6 h (Fig 3C, right and middle panels). Resection may have never reached the 15.2-kb region in *SRS2* (the values at all time points are close to 1), perhaps due to completion of re-synthesis before resection has reached the region (Fig 3C, left panel). At the same time, only a small fraction of *srs2Δ* mutants possessed dsDNA in this region by the end of the experiment (6-h time point). Therefore, Srs2 is required for re-synthesis of resected DNA during BIR.

Break-induced replication in our system results in addition of ~94-kb sequence from chr.IIR to the DSB site (Fig 3A). Since ~60% of wild-type cells successfully repair DSBs by BIR, in the post-repair population the relative amount of DNA corresponding to the 94-kb sequence should increase 1.6-fold (100% on chr.IIR + 60% copied to chr.VIIL). Progression of BIR was monitored by Southern blotting using BIR6, BIR36 and BIR77 probes corresponding to DNA

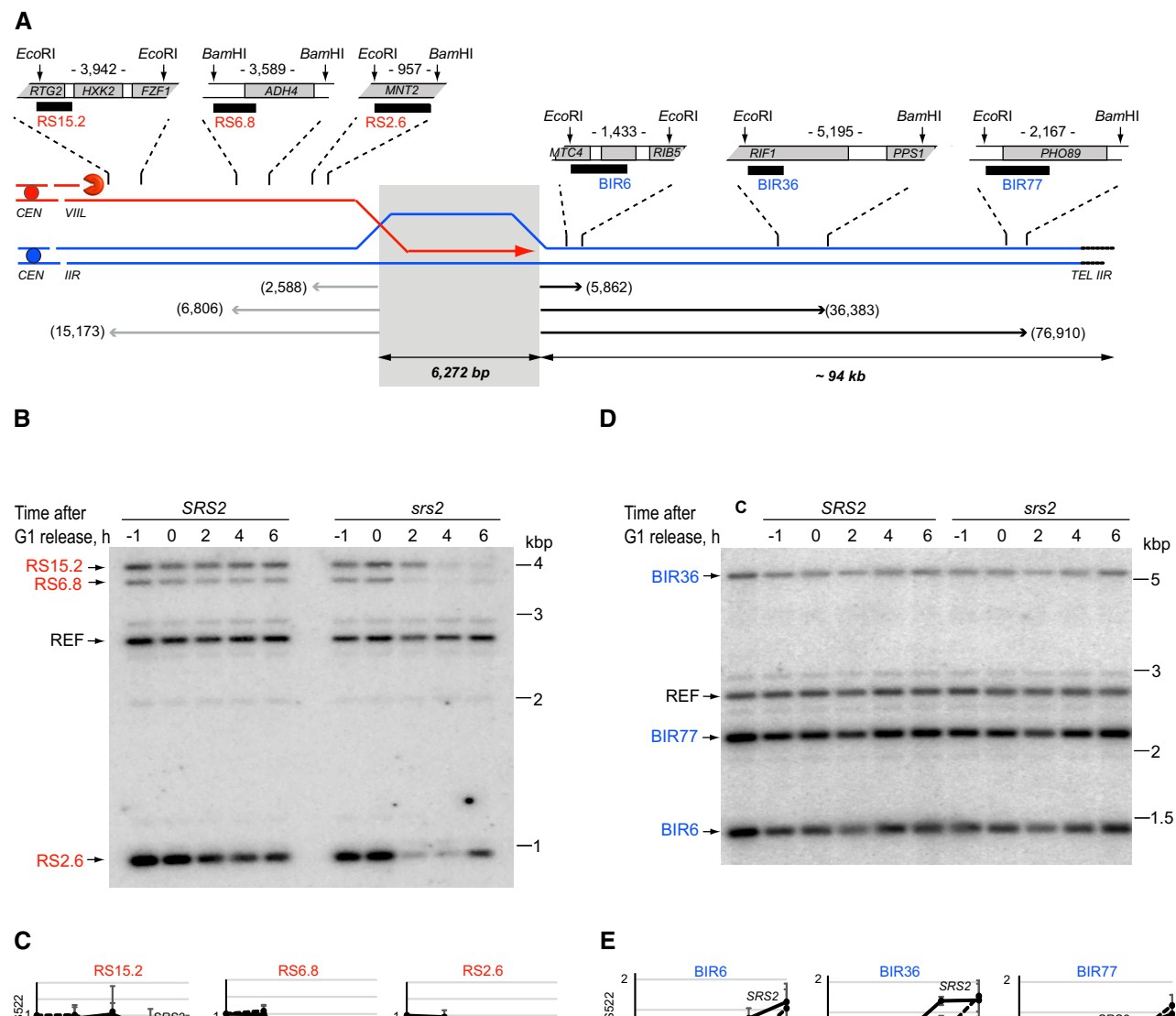

Figure 3.  Analysis of Srs2 requirement in BIR.

A   Schematic of the quantitative BIR assay. Modified chr.VIIL (red) and chr.IIR (blue) share a 6,272-bp homology (grey shadow) used to repair an HO-induced DSB by BIR. Black boxes indicate hybridization probes used in Southern blotting experiments to monitor re-synthesis of resected DNA (RS probes, red) and BIR (BIR probes, blue), respectively. Numbers next to the one-ended arrows indicate distances (in bp) from the homology to the distal restriction sites of the DNA fragments analysed by Southern blotting using the corresponding probes. Numbers between the restriction sites indicate the sizes of restriction fragments detected by the corresponding hybridization probes.

B   Southern blot analysis of re-synthesis of resected DNA during BIR in *SRS2* and *srs2Δ* corresponding to the data quantifications in (C). DNA was digested with *Eco*RI and *Bam*HI, resolved on 0.7% agarose gels, transferred onto charged Nylon membrane and hybridized to the mixture of four probes (three RS probes and a reference probe, REF, hybridizing to an *ARS522*-containing fragment on chr.V which is not involved in the repair). A representative image of one of the three repeats is shown.

C   Re-synthesis of resected DNA on chr.VIIL in *SRS2* and *srs2Δ* cells (solid and dashed lines, respectively) at the distance of 15.2 (RS15.2), 6.8 (RS6.8) and 2.6 (RS2.6) kb away from the homology region. Average ± SD (*n* = 3) is shown for each time point.

D   Southern blot analysis of BIR-dependent DNA synthesis in *SRS2* and *srs2Δ* corresponding to the data quantifications in (E). DNA was digested with *Eco*RI and *Bam*HI, resolved on 0.7% agarose gels, transferred onto charged Nylon membrane and hybridized to the mixture of four probes (three BIR probes and a reference probe, REF, hybridizing to an *ARS522*-containing fragment on chr.V which is not involved in the repair). A representative image of one of the three repeats is shown. C indicates control strain NK3980.

E   BIR-dependent DNA synthesis in *SRS2* and *srs2Δ* cells (solid and dashed lines, respectively) at the distance of 6 (BIR6), 36 (BIR36) and 77 (BIR77) kb away from the homology region. Average ± SD (*n* = 3) is shown for each time point.

Data information: Strains used: NK4070, NK4079; NK5321, NK5322.

sequences located 6, 36 and 77 kb away from the homology region, respectively. The *srs2Δ* mutation resulted in slower but successful BIR-dependent DNA synthesis: like wild-type cells, *srs2Δ* reached 1.6-fold increase in chr.IIR sequences by the end of the time-course experiments (Fig 3D and E). Therefore, during BIR, Srs2 is predominantly required for restoring resected DNA.

### Srs2-dependent removal of Rad51 is necessary for efficient DNA synthesis during SSA

To investigate the effect of *srs2Δ* on SSA, we used a genetic system where *ura3-52* and *URA3* were separated by ~4 kb of DNA which included *KAN* and a recognition site for the HO-nuclease expressed from a galactose-inducible promoter (Fig 4A). *SRS2* and *srs2Δ* cells pre-grown on YP + raffinose agar were plated on YPD (to score total cell titre in the experiment) and YP + galactose plates for DSB induction. On galactose, upon DSB repair via SSA the vast majority of cells become Kan^s Ura^−, as the 766-bp homology closest to the break in *ura3-52* is predominantly used. The ratio between the Kan^s Ura^− colonies grown on galactose plates and the ones on YPD was used to calculate the frequency of SSA (Fig 4B). Consistent with the previously published results (Vaze *et al*, 2002), *srs2Δ* conferred a genetic defect in SSA which was suppressed by a deletion of *RAD51* (Fig 4B).

Single-strand annealing involves (i) DSB processing to generate ssDNA at the regions of homology, (ii) annealing of the homologous sequences, (iii) Rad1/Rad10-dependent cleavage of the non-homologous ssDNA ends, and (iv) DNA synthesis to reconstitute DNA integrity at the repair loci (Symington *et al*, 2014) (Fig EV1A). To determine whether Srs2 was required at any of the first three steps, we monitored the cleavage of non-homologous ends using qPCR spanning the cleavage point but observed no significant effect of *srs2Δ* on the progress of SSA at this stage (Fig EV1B–E). Therefore, Srs2 loss has no effect on DSB resection (at least up to the processing of the homologous regions), annealing of the ssDNA homologies, or Rad1/Rad10-dependent cleavage of non-homologous ends and the defect of *srs2Δ* mutants in SSA should be attributed to a later stage of repair.

*SRS2* and *srs2Δ* might differ either in DSB resection over longer distances or in DNA repair synthesis required to complete SSA. To compare resection in *SRS2* and *srs2Δ*, a pair of isogenic strains with unrepairable DSBs was constructed by removing the *ura3-52* allele from the yeast used for SSA assays (Fig EV2A). Three different probes, R5, R14 and R20, specific to DNA sequences 5, 14 and 20 kb away from the break, respectively, were used to monitor DSB resection over time by Southern blotting (Fig EV2A and B). *SRS2* and *srs2Δ* showed very similar behaviour in break processing (Fig EV2C), and therefore, Srs2 is not involved in DSB resection.

To assay the dynamics of DNA synthesis during SSA, we analysed the progress of *Sal*I-*Eco*RI fragment formation (fragment L—Long) which involved a total of 7.9 kb of DNA synthesis to reach the restriction sites (Fig 4C). Consistent with the genetic data (Fig 4B), *srs2Δ* led to slower L-fragment generation, whereas loss of Rad51 resulted in the suppression of the *srs2Δ* defect on the rate of the fragment L formation (Figs 4D and EV3). We next tested the requirement of Rad51 removal by Srs2 for DNA synthesis in our reconstituted *in vitro* strand extension assay using purified proteins (Sebesta *et al*, 2011) and observed robust Rad51-dependent

synthesis inhibition which was almost fully suppressed by Srs2 (Fig 4E and G). Therefore, presence of Rad51 inhibits DNA synthesis and its removal by Srs2 alleviates this inhibition.

### Removal of Rad51 by Srs2 is required for PCNA loading onto DNA

To gain insights into the mechanisms of DNA synthesis inhibition by Rad51 *in vivo*, two shorter DNA fragments of different lengths (fragments S1 and S2) were monitored through SSA by quantitative Southern blotting (Figs 5A and B, and EV4A and B). The fragment S1 required a minimum of 14 and 4 bp of DNA synthesis to produce dsDNA at the *BspCN*I and *Sma*I sites, respectively (Fig 5A). The fragment S2 required 358 and 1,396 bp of DNA synthesis to reach the two *Bgl*II sites on either side of homology (Fig 5A). Comparative analysis of fragments S1 and S2 in *SRS2* and *srs2Δ* revealed a significant difference between the two strains, with *srs2Δ* showing ~0.5 h delay in production of both dsDNA fragments (Fig 4B, red arrow). However, there was no significant difference between the production of the two fragments within either *SRS2* or *srs2Δ* (Fig 5B) although generation of the fragment S2 involved ~100 times more DNA synthesis than the fragment S1 did (14 + 4 vs. 358 + 1,396 bp). This highly important observation suggests that the slower fragment generation in the *srs2Δ* mutants could not be attributed to a slower rate of DNA polymerization *per se* because if this were the case then the *srs2Δ* mutant strain would have shown a drastic difference between the fragments S1 and S2 due to significantly more DNA synthesis (~100-fold) involved in the production of the fragment S2. In fact, it proves the opposite: the rate of DNA synthesis *per se* is so fast in both the wild-type and mutant cells that the 100-fold length difference in the analysed DNA synthesis tracts S1 and S2 cannot be differentiated in our experimental setting. Therefore, the observed difference between the *SRS2* and *srs2Δ* cells in the generation of the fragments S1 and S2 must be due to a step in repair which takes place in the narrow window after the non-homologous end cleavage by Rad1/Rad10, but prior to the start of DNA synthesis (Fig EV1) and this step must be slower in the *srs2Δ* mutants. Such step is likely to be the recruitment of the DNA synthesis machinery to the repair locus, in particular RFC-PCNA which requires the presence of RPA at the recruitment loci (Yuzhakov *et al*, 1999) and which in turn could be abrogated by the presence of Rad51 at the potential PCNA recruitment site.

Our attempts to assay PCNA recruitment to the repair loci *in vivo* by ChIP were unsuccessful, perhaps, because PCNA is present there very transiently (it would move away along with the replication machinery as soon as polymerase is recruited), irrespectively of whether PCNA recruitment is fast or has a delay. To test the effect of Rad51 on PCNA loading *in vitro*, we radio-labelled PCNA and monitored its loading on DNA substrate using PCNA-loading assay (Fig 5C). As reported earlier (Yuzhakov *et al*, 1999), RPA greatly stimulated the loading of PCNA (Fig 5D, lanes 2 and 3). In contrast, Rad51 dramatically inhibited PCNA loading (Fig 5D, lanes 3–7), but addition of increasing concentrations of the Srs2 fragment 1–910 resulted in suppression of the Rad51 inhibitory effect (Fig 5E and G), thereby confirming the requirement of Rad51 removal prior to loading of PCNA and initiation of DNA repair synthesis. The ability to counteract Rad51 was specific to Srs2 as Pif1 helicase could not substitute for Srs2 (Fig 5H). If Srs2 removes Rad51 so that it could be replaced on ssDNA by RPA, then higher RPA concentration in

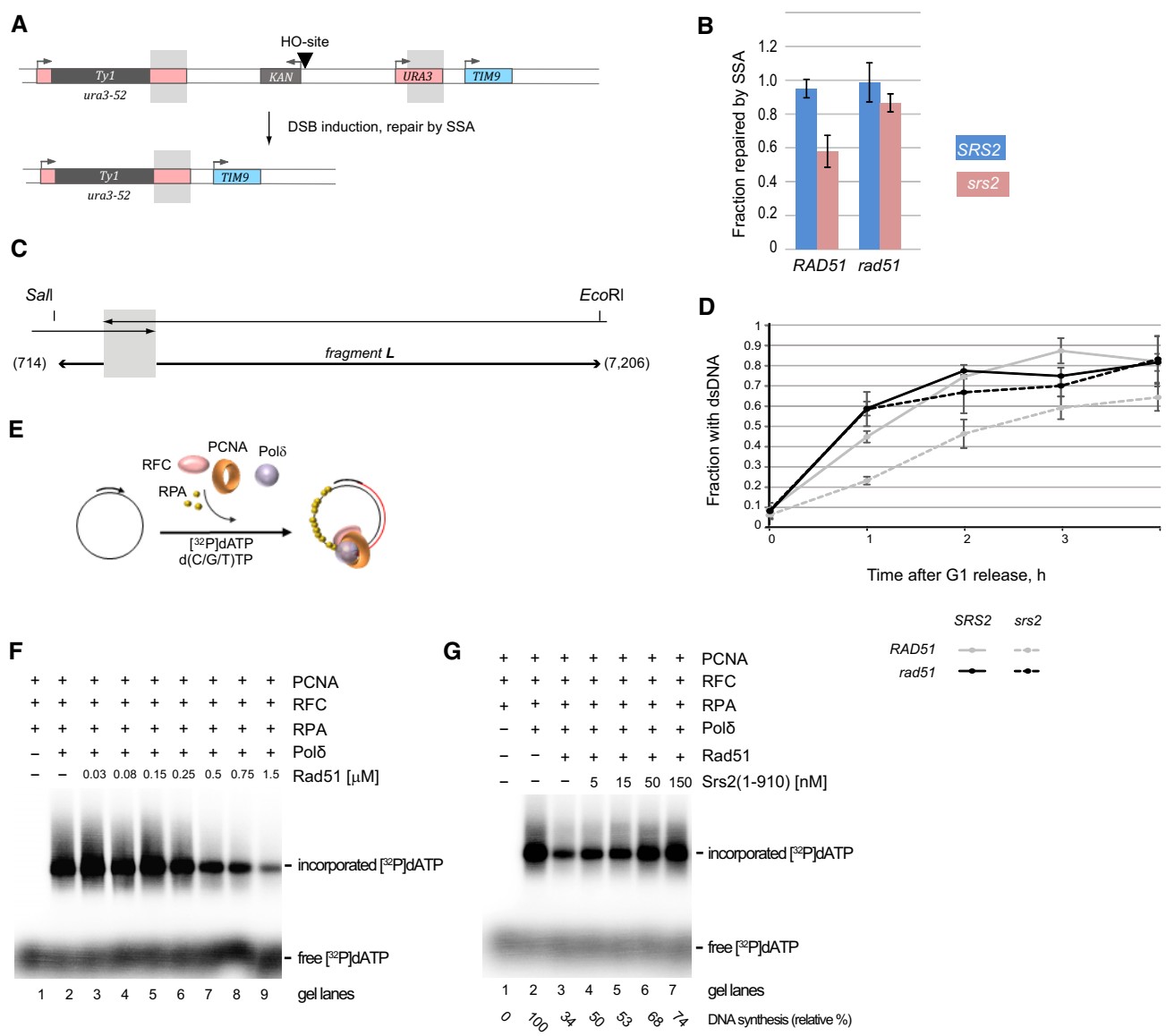

**Figure 4. Srs2 is required to relieve Rad51-dependent inhibition of DNA synthesis.**

A  Schematic of the genetic system used to analyse inducible DSB repaired by SSA. Chr.V contains *ura3-52* and *URA3* (at the endogenous *URA3* locus) separated by ~4 kb of DNA containing *KAN* (grey box, arrow above indicates the promoter) and an HO site (triangle). Galactose-inducible expression of the HO endonuclease leads to DSB formation at the HO site. After DSB repair via SSA, the majority of cells become Kan^s Ura^− as the 766-bp homology between *URA3* and *ura3-52* (grey shadows) is predominantly used.

B  Frequency of DSB repair via SSA in *SRS2* and *srs2Δ* cells with and without functional HR in the assay based on the system shown in (A). Average ± SD (*n* = 4) is shown for each genotype. Strains used: NK4691–4693; NK4805–4808; NK5081–5084; NK5085–5091.

C  Schematic of the quantitative SSA assay used in panel (D). Grey shadow represents the annealing region of 766 bp present on both sides of a DSB. Numbers next to the one-ended arrows indicate distances (in bp) from the homology to the restriction sites, *Sal*I and *Eco*RI, used to generate DNA fragment L analysed by Southern blotting.

D  Fragment L formation in *SRS2* and *srs2Δ* cells in the presence and absence of Rad51. See also Fig EV3 for blot images. Average ± SD (*n* = 4) is shown for each time point. Strains used: NK4691–4693; NK4805–4808; NK5081–5084; NK5085–5091.

E  Schematic for the basic DNA strand extension assay used in (F and G).

F  Rad51 inhibits DNA synthesis of φX174 ssDNA substrate (0.5 nM) by Polδ. Increasing amount of Rad51 (0.03, 0.08, 0.15, 0.25, 0.5, 0.75, 1.5 μM) was incubated with the pre-loaded replication complex (RFC (17.5 nM), proliferating cell nuclear antigen (PCNA) (10 nM) and DNA) and DNA synthesis was started by the addition of Polδ (10 nM) and nucleotides containing α-³²P labelled dATP.

G  Srs2(1–910) suppresses the inhibition of DNA synthesis by Rad51. The reaction was carried out the same way as in (F) except of the increasing amount of Srs2(1–910) (5, 15, 50, 150 nM) was added to indicated reactions before the start of DNA synthesis. The relative % of DNA synthesis is indicated.

the system should increase RPA occupancy on DNA and suppress the need for Srs2. Indeed, raising RPA concentration from 0.08 to 1.4 μM resulted in a significant increase in PCNA loading in the presence of the same amounts of Rad51 and almost completely suppressed Rad51-dependent inhibition of this process (Fig 5I). These results can be explained by mutually exclusive binding of

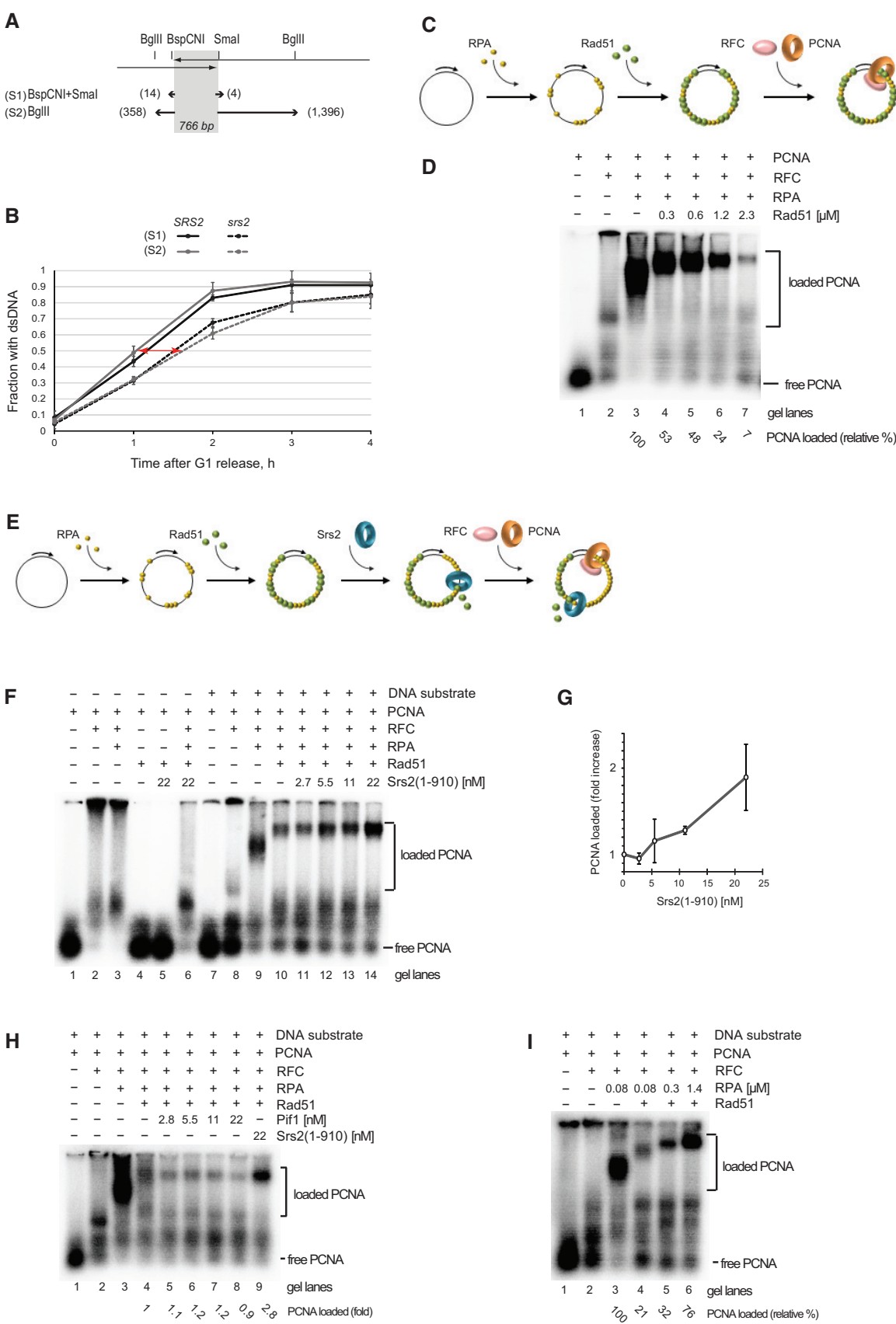

Figure 5.

**Figure 5. Srs2-dependent removal of Rad51 is required for efficient PCNA loading.**

A   Schematic of the quantitative SSA assay used in (B). Grey shadow represents the annealing region of 766 bp present on both sides of a DSB. Numbers next to the one-ended arrows indicate distances (in bp) from the homology to the restriction sites used to generate DNA fragments analysed by Southern blotting: *Bsp*CNI and *Sma*I for the fragment S1, and *Bgl*II for the fragment S2.

B   Generation of double-stranded fragments S1 and S2 in *SRS2* and *srs2Δ* during a time-course experiment. See also Fig EV2B and C for blot images. Red two-ended arrow indicates time delay of ~0.5 h in fragment S1/S2 formation in *srs2* mutants relative to *SRS2* cells. Average ± SD (*n* = 4) is shown for each time point. Strains used: NK4691–4693; NK4805–4808.

C   Schematic of PCNA-loading assay used in (D and I).

D   Rad51 inhibits PCNA loading. Increasing amount of Rad51 (0.3, 0.6, 1.2, 2.3 μM) was added to φX174 ssDNA substrate (0.5 nM) pre-incubated with RPA (75 nM). Loading of PCNA was started by the addition of RFC (21 nM) and $^{32}$P-PCNA (10 nM). The relative amount of loaded PCNA in each reaction is indicated below.

E   Schematic of PCNA-loading assay shown in (F and H).

F   Srs2(1–910) overcomes the inhibitory effect of Rad51 on PCNA loading. φX174 substrate (0.5 nM) was pre-incubated with RPA (75 nM) followed by the addition of Rad51 (2.3 μM). Increasing amounts of Srs2(1–910) (2.7, 5.5, 11, 22 nM) were added to the reaction, and PCNA loading was started by the addition of RFC (21 nM) and $^{32}$P-PCNA (10 nM).

G   Quantitative analysis of PCNA loading as a function of increased Srs2(1–910) concentration. Average ± SD (*n* = 3) is shown for each Srs2 concentration.

H   Pif1 cannot substitute for Srs2 in promoting PCNA loading. Experiments were done as in (F), except Pif1 at 2.8, 5.5, 11 and 22 nM was used instead of Srs2 in samples shown in lanes 5–8. Lane 9, control reaction with 22 nM Srs2(1–910). The relative amount of loaded PCNA in each reaction is indicated below.

I   Increased concentrations of RPA suppress the inhibitory effect of Rad51 on PCNA loading. φX174 substrate (0.5 nM) was pre-incubated with RPA (0.08, 0.3 and 1.4 μM) followed by the addition of Rad51 (2.3 μM). PCNA loading was started by the addition of RFC (21 nM) and $^{32}$P-PCNA (10 nM).

ssDNA by RPA and Rad51 whereby Srs2-dependent removal of Rad51 indirectly promotes RPA binding and subsequent PCNA loading onto DNA.

Comparative analysis of DNA synthesis during SSA involving all the three fragments described above (Fig 6A) revealed that in *srs2Δ* mutants fragment L was restored ~0.5 h later than S1/S2 (Fig 6B, blue arrow). Because the fragment S1 vs. S2 comparison (Fig 5B) suggests indistinguishably fast rate of DNA synthesis in both *SRS2* and *srs2Δ* strains and proves that the difference between *SRS2* and *srs2Δ* is not due to slower DNA polymerization *per se*, the difference between the fragments S2 and L in the *srs2Δ* cells has to be attributed to additional Srs2-dependent events taking place on longer

DNA tracts. These events could be additional rounds of PCNA loading on longer DNA, possibly due to spontaneous disruptions of DNA synthesis followed by stalling and/or disassembly of the replication machinery.

### Relative rates of DNA resection and re-synthesis of processed DNA are important for completion of DSB repair

We hypothesized that break resection would normally be chased and terminated by DNA re-synthesis which would have a much faster rate than the processing [140 kb/h for conventional replication (Raghuraman *et al*, 2001), but could be slower during repair

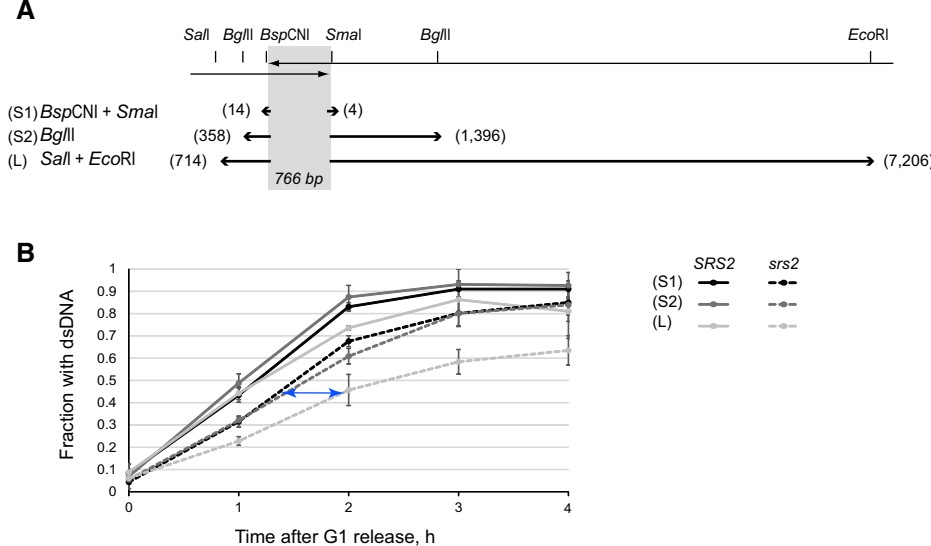

**Figure 6. Using an increased length DNA fragment to monitor re-synthesis of processed DNA during SSA.**

A   Schematic of DNA repair involving generation of the fragments S1 (*Bsp*CNI-*Sma*I), S2 (*Bgl*II-*Bgl*II) and L (*Sal*I-*Eco*RI) as products of SSA (see Figs 4A and 5A for further explanation).

B   Comparative analysis of fragment S1, S2 and L production in *SRS2* and *srs2Δ* cells during a time-course experiment (combined data from Figs 4D and 5B). Blue two-ended arrow indicates a ~0.5 h time difference between the formation of fragments S2 and L in *srs2Δ* mutants. Since the distance between the *Bgl*II site on the right and the *Eco*RI site is ~6 kb, the rate of restoration of dsDNA in *srs2Δ* mutants can be roughly estimated at 12 kb/h. The rate of re-synthesis is much faster in wild-type cells as the difference between S2 and L is not even detectable at the 1 h time point. Strains used: NK4691–4693; NK4805–4808. Average ± SD (*n* = 3) is shown for each time point of each genotype.

synthesis, vs. 4 kb/h for resection (Fishman-Lobell *et al*, 1992)]. In fact, by comparing the reconstitution of fragments S2 and L (Fig 6B), we could estimate the rate of dsDNA restoration in *srs2Δ*. The fragments have relatively short DNA stretches to synthesize on the left-hand side of the homology (*Sal*I side), and therefore, the full fragment reconstitution is likely to be dictated by DNA synthesis on the other side (*Eco*RI side) (Fig 6A). DNA synthesis at the *Eco*RI site occurs ~0.5 h after the restoration of the *Bgl*II site which is 5,810 bp away from the *Eco*RI site (Fig 6B, blue arrow). This means that in *srs2Δ*, dsDNA is restored with the average rate of ~12 kb/h, but it consists of phases of fast movement of replication machinery interrupted by slow recruitment of RFC/PCNA due to Rad51 presence. Because resection always has a head-start over DNA re-synthesis and in the absence of Srs2 re-synthesis is impeded by slow PCNA recruitment, the polymerase in some of the *srs2Δ* cells may never catch the "run-away" resection machinery. This explains why *srs2Δ* mutants accumulate ssDNA, often far away from the damage site, cannot inactivate the DNA damage checkpoint and die (Yeung & Durocher, 2011). Therefore, Rad51 removal by Srs2 might be required for efficient PCNA loading not only at the site of initiation of DNA synthesis but along the whole length of processed DNA before it can be restored to its double-stranded form.

According to our hypothesis above, the following predictions can be made. Further slowing down of DNA re-synthesis might exacerbate the effect of *srs2Δ* on cell survival, whereas slowing down DNA resection would have the opposite effect by helping *srs2Δ* mutants to complete repair. Consistent with this hypothesis, loss of the DSB resection nuclease Exo1 partially compensated for the lack of Srs2 during SSA and completely suppressed the *srs2Δ* defect in BIR and *de novo* telomere addition (Fig 7A). To slow down DNA synthesis during SSA, we used low concentrations of hydroxyurea (HU at 5, 10 and 25 mM) to deplete dNTPs pool. In the genetic assays, the drug had no effect on SSA in *SRS2* cells but further reduced the survival of *srs2Δ* yeast in a concentration-dependent manner (Fig 7B). The reconstitution of the fragment L in the presence of 25 mM HU was delayed in both strains, likely due to a slower rate of nucleotide incorporation in the presence of HU. While the wild-type cells efficiently reconstituted the fragment in the presence of HU, albeit with a 1 h delay, *srs2Δ* could not re-synthesize the fragment L with the same efficiency as the cells repairing the break in the absence of HU (Fig 7C). Therefore, slowed down DNA re-synthesis further reduces the efficiency of SSA in *srs2Δ* mutants, while *exo1Δ* suppresses the defect of *srs2Δ* in DSB repair.

### The role of Srs2 in DSB repair is different from its role at replication forks

At the C-terminus, Srs2 contains a variety of regulatory motifs, which are modulated through post-translational modifications and/or required for the interactions of Srs2 with other proteins, including PCNA, and these are important for its role at replication forks (Papouli *et al*, 2005; Pfander *et al*, 2005; Burgess *et al*, 2009) and regulation of the D-loop extension (Burkovics *et al*, 2013). However, most of the C-terminus was not required for the role of Srs2 in DSB repair via *de novo* telomere addition, BIR and SSA (Fig 8A–D). Longer C-terminal truncations including BRCv motif required for Srs2–Rad51 interactions resulted in a partial

loss of Srs2 activity both *in vivo* (*de novo* telomere addition) and *in vitro* (Fig 8C and E). In contrast, the Srs2 ATPase activity was important for all the repair mechanisms analysed (Fig 8B–D). Therefore, the role of Srs2 in DSB repair is different from its role at replication forks and does not require Srs2–PCNA interaction (Papouli *et al*, 2005; Pfander *et al*, 2005). Instead, Srs2 acts upstream of PCNA by removing Rad51 from DNA repair loci in order to stimulate PCNA recruitment, thereby promoting the speed with which ssDNA is converted into its functional double-stranded form.

## Discussion

Homology-dependent DSB repair mechanisms require extensive resection of broken ends which are then used as a platform for localization of DNA damage signalling complexes as well as DNA repair machineries, in order to trigger DNA damage checkpoint, cell cycle arrest and break repair (Symington *et al*, 2014; Villa *et al*, 2016). At the late stages of repair, ssDNA must be restored into a double-stranded state by re-synthesis of the resected DNA which will otherwise constantly signal for checkpoint activation leading to a persistent cell cycle arrest and cell death. Failure to complete DNA re-synthesis step might also be fatal for the cell due to the inability to transcribe genes at resected DNA loci (Manfrini *et al*, 2015). In spite of its importance for the completion of repair, DNA re-synthesis remains poorly understood, particularly in comparison with the earlier steps of DSB repair. Here we show that the Srs2-dependent removal of Rad51 from resected DNA promotes restoration of dsDNA: Rad51 dislodging from ssDNA allows its replacement with RPA which in turn recruits RFC-PCNA and promotes PCNA loading and subsequent initiation of re-synthesis of resected DNA.

In this study, we used quantitative analyses to monitor progression of DSB repair by multiple mechanisms and demonstrated that *srs2Δ* mutants have a defect in re-synthesis of resected DNA. This defect is caused by the presence of Rad51 on resected DNA as loss of *RAD51* eliminates the need for Srs2 during SSA and *de novo* telomere addition. Moreover, Rad51 inhibits DNA re-synthesis in wild-type cells as even in the presence of Srs2 SSA products form faster in *rad51Δ* than in *RAD51* (Fig 4D). Localization of Rad51 to ssDNA does not interfere with the *in vivo* DNA synthesis *per se* but rather limits PCNA loading onto DNA as shown in our *in vitro* experiments. PCNA recruitment to DNA relies on RFC–RPA interaction specifically at the ss-dsDNA junction (Yuzhakov *et al*, 1999). Rad51 bound at the junction is not a suitable substrate for RFC interaction, and therefore, Srs2 is needed to dislodge Rad51 from ssDNA in order to allow RPA binding and PCNA loading. Consistent with this model, excess of RPA which can outcompete Rad51 for DNA binding partially compensates for the need of Srs2 in our *in vitro* PCNA-loading assays (Fig 5I).

DNA re-synthesis during BIR and *de novo* telomere addition, but not SSA, requires Polα primase to initiate synthesis of the complementary strand. Recruitment of Polα to DNA and its activity which both rely on Polα interaction with RPA (Braun *et al*, 1997) might also depend on Rad51 removal by Srs2. During *de novo* telomere addition, the primase is expected to be recruited to the newly added TG repeats, perhaps via Pol12–Stn1 interaction (Grossi *et al*, 2004). It remains unclear whether Rad51 filaments would spread into the

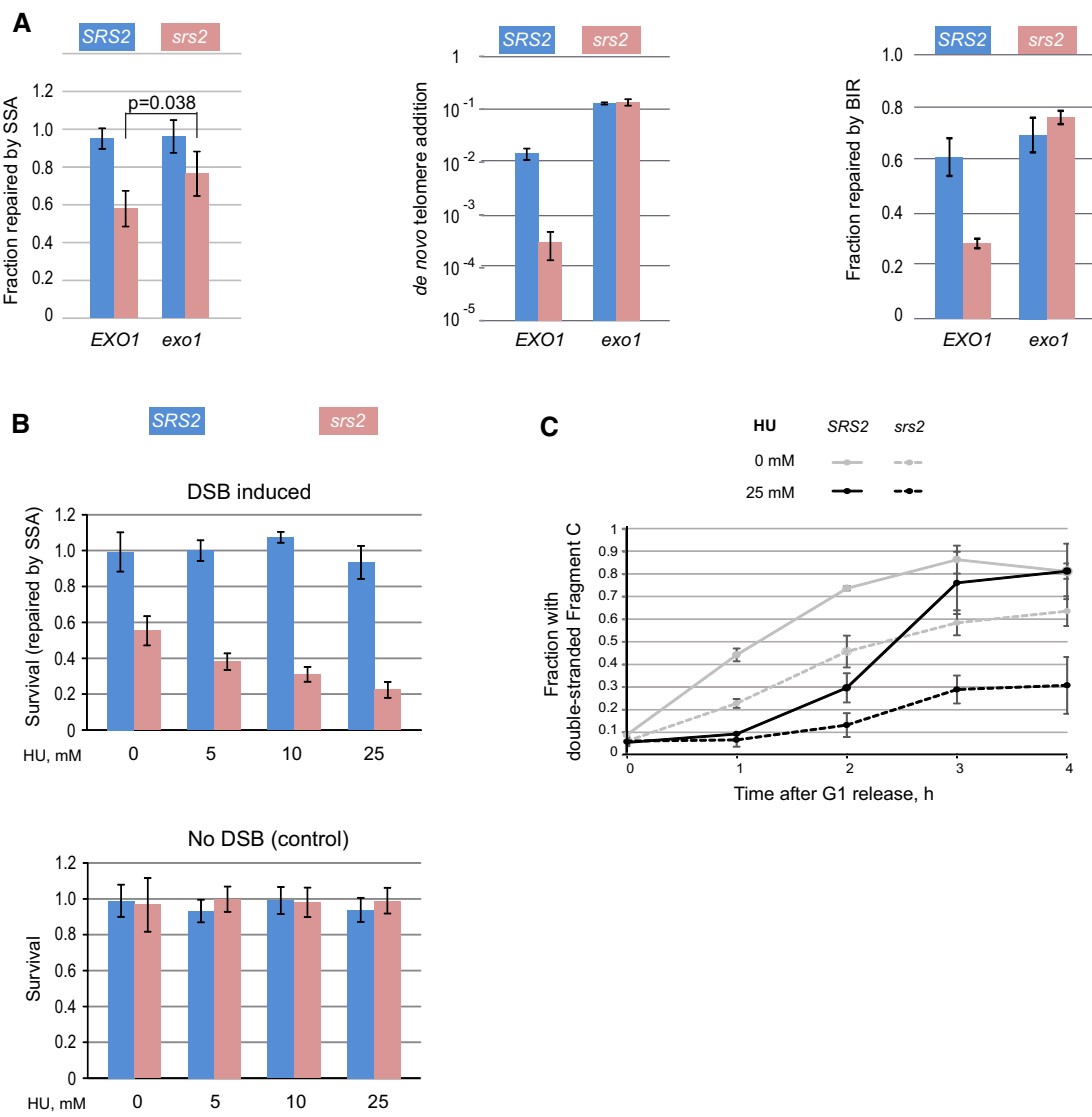

**Figure 7. The *srs2Δ* defects in DNA repair can be suppressed by slowing down resection and exacerbated by slowing down DNA synthesis.**

A   Loss of Exo1 suppresses *srs2Δ* defect in multiple DSB repair mechanisms involving extended resection of broken ends: SSA (left), *de novo* telomere addition (middle) and BIR (right). Average ± SD (*n* = 4) is shown for each genotype for all experiments presented. Unpaired *t*-test was used to calculate the *P*-value shown for the SSA experiments. Strains used: SSA: NK4691–NK4693; NK4805–4808; NK5070–5073; NK5074–5080; *de novo* telomere addition: NK1264; NK2375, NK2376, NK2016, NK2017; NK5244, NK5245; BIR: NK4070, NK4079; NK5321, NK5322; NK5446, NK5447; NK5448, NK5449.

B   The effect of HU on the survival of *SRS2* and *srs2Δ* cells after DSB induction (top panel). The control experiment involving similar HU treatments in the absence of DSBs is shown in the bottom panel. Average ± SD (*n* = 4) is shown for each genotype. Strains used: NK4691–4693; NK4805–4808.

C   Formation of fragment L in *SRS2* and *srs2Δ* cells in the presence and absence of 25 mM HU. Average ± SD (*n* = 3) is shown for each time point. Strains used: NK4691–4693; NK4805–4808.

telomeric sequences or if Cdc13-Stn1-Ten1 would prevent the spreading.

Lack of Srs2 does not block the re-synthesis completely: consistent with the previously published results (Vaze *et al*, 2002) products of DSB repair can be observed by Southern blotting in *srs2Δ* cells. However, quantitative analysis of repair progression shows that *srs2Δ* cells re-synthesize resected DNA at a much slower rate than wild-type yeast. We believe that efficient re-synthesis has a dual function in DNA repair. Firstly, it restores resected DNA into its original double-stranded form, and secondly, it is required to terminate further resection. While the inefficient re-synthesis in the absence of Srs2 can only partially perform the first role, as a result of it, it often fails at the second one resulting in a defect in DSB repair in *srs2Δ* mutants. Although in the mutant cells, repair can be completed around the break site, albeit with a delay, ssDNA gaps "migrate" further away from DSBs (Fig 3), consistent with the previously observed accumulation of ssDNA long distance away from the initial damage site (Yeung & Durocher, 2011). Our understanding of the role of Srs2 in DNA repair is also supported by the previously published data on SSA in a set of three strains

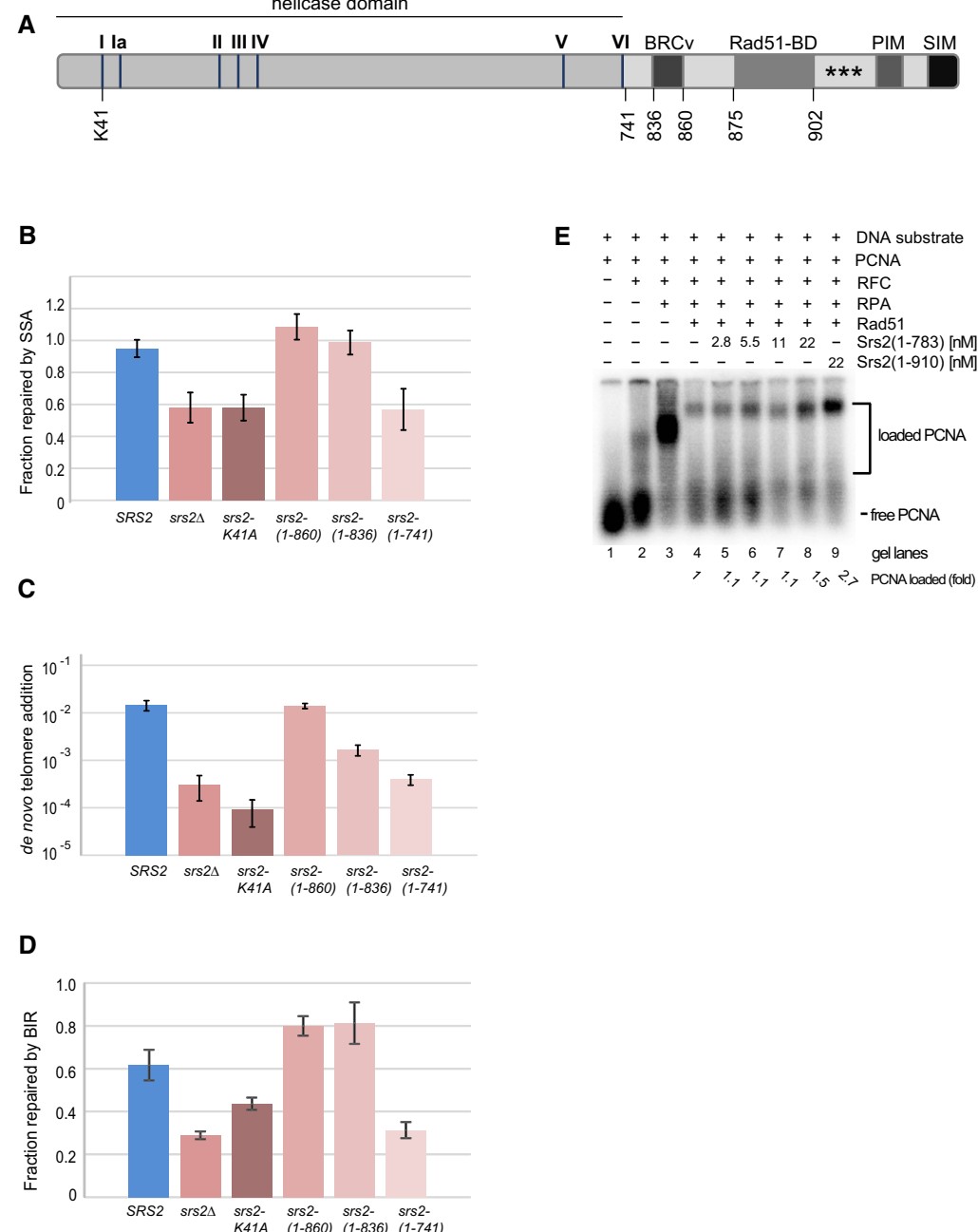

**Figure 8. The role of Srs2 in DSB repair requires its ATPase activity but is independent of its C-terminus.**

A    Schematic of the full-length Srs2 protein shown as a bar. I–VI, Srs2 helicase motifs; K41, a lysine residue required for ATP binding and hydrolysis; BRCv, BRC repeat variant motif; Rad51-BD, Rad51-binding domain; PIM, PCNA-interacting motif; SIM, SUMO-interacting motif; asterisks indicate sumoylation sites. Numbers below the protein indicate positions of amino acid residues within the Srs2 protein.

B–D    The efficiency of DSB repair via SSA (B), *de novo* telomere addition (C) and BIR (D) in different alleles of *SRS2*. Average ± SD (*n* = 3) is shown for each genotype in all experiments. Strains used: SSA: NK4691–4693; NK4805–4808; NK5104–5107; NK5066–5069; NK5062–5065; NK5058–5061; *de novo* telomere addition: NK1264; NK2375, NK2376, NK3332–3334; NK3308–3310; NK4217, NK4247; NK3353–3355; BIR: NK4070, NK4079; NK5321, NK5322; NK5536, NK5537; NK5450, NK5451; NK5452, NK5453; NK5454, NK5455.

E    Srs2(1–783) is less active than Srs2(1–910) in promoting PCNA loading *in vitro*. The assay was performed as shown in Fig 5E.

which differ in the distance between the homologies: 0.7, 5 and 30 kb (Vaze *et al*, 2002). The *SRS2* deletion conferred the strongest defect in SSA in the background with the longest distance between the homologies by bringing down the DSB survival rates to 55, 10

and < 2%, respectively (Vaze *et al*, 2002). Since the position of the break was next to one of the homologies and in turn 0.7, 5 or 30 kb away from the other one, by the time the farthest from the break homology was processed (so that both homologies are

available for the annealing step), there was ~0.7, 5 and 30 kb, respectively, of to-be-restored ssDNA generated on the other side of the break. Since re-synthesis is impaired in *srs2Δ*, having resection machinery which is already 30 kb away from the start of the re-synthesis locus presents a much harder problem than if this distance equals to only 0.7 or 5 kb.

We propose a model where the fast rate of re-synthesis of processed DNA is necessary for the replication machinery to catch nucleases in order to stop further resection and restore strand continuity (Fig 9). In the absence of Srs2, the recruitment of PCNA is impaired and DNA restoration is slowed down. When re-synthesis involves long tracts of ssDNA, the replication machinery is likely to undergo multiple rounds of disassembly–reassembly and the role of Srs2 in Rad51 removal to stimulate PCNA recruitment becomes critical for successful restoration of dsDNA. In *srs2Δ*, the replication machinery often fails to catch the "run-away" resection and cells accumulate ssDNA as a result of unsuccessful repair.

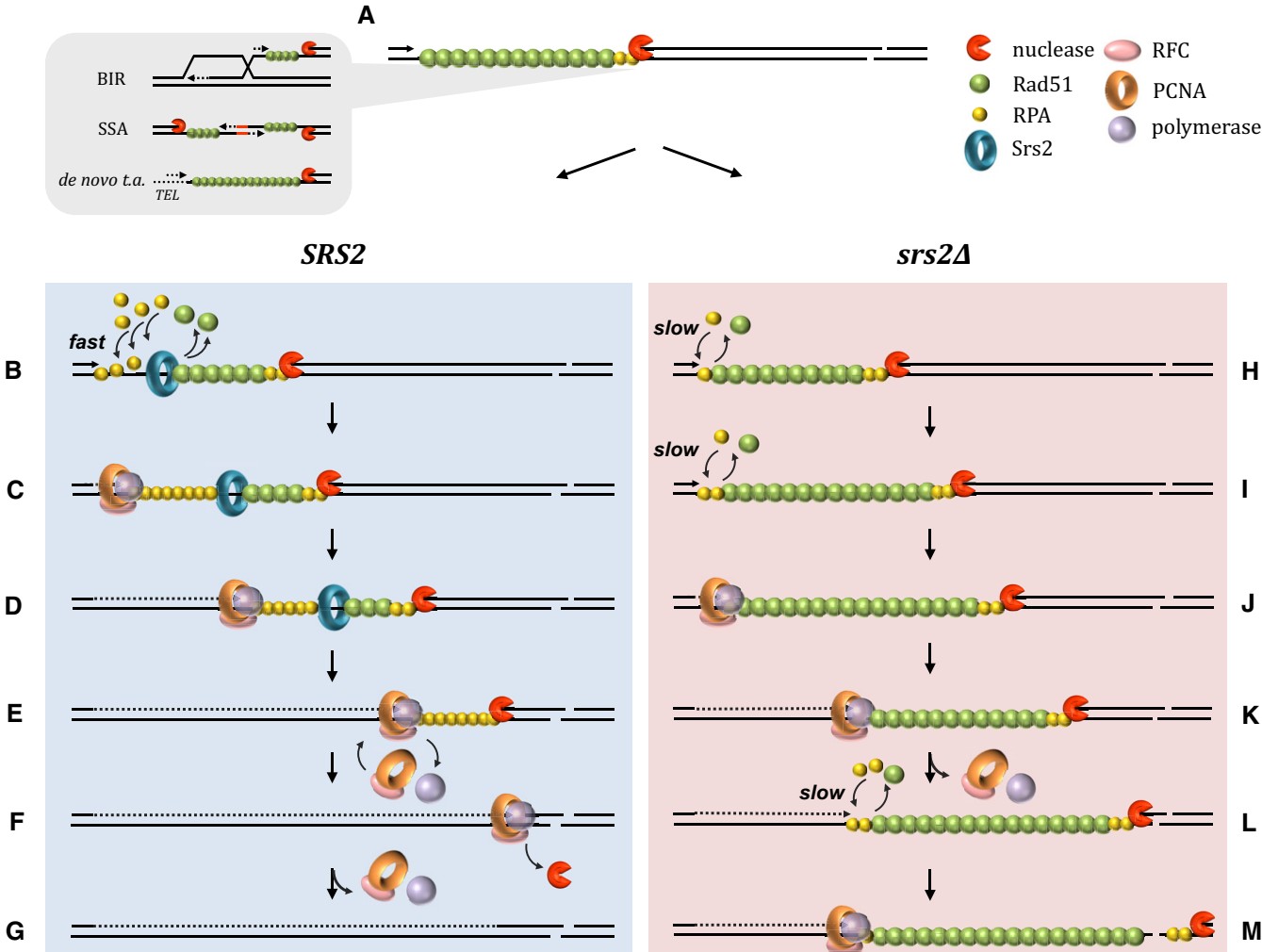

**Figure 9.  A model for the role of the Srs2 helicase in restoration of dsDNA during DSB repair.**
During DSB repair by a variety of mechanisms shown on the left, ssDNA gaps require DNA synthesis to restore dsDNA (a). ssDNA formed as a result of DSB processing is covered by Rad51. In wild-type cells, the Srs2 helicase displaces Rad51 (b), thereby promoting RPA binding to the ssDNA. Presence of RPA at ssDNA-dsDNA junction is required for PCNA loading (via RPA–RFC interaction) followed by recruitment of DNA polymerase (c). Because DNA synthesis is faster than resection, the ssDNA gap shortens (d). When long stretches of DNA have to be synthesized, the likelihood of the replication machinery stalling and disassembly is increased (e). However, its re-assembly is efficient in wild-type cells as Srs2 ensures that ssDNA is clear from Rad51 and covered with RPA. Once loaded, the polymerase rapidly catches up with the resection machinery due to a higher speed of replication vs. resection (f). As a result, DNA processing is terminated. Therefore, due to efficient initial loading as well as re-loading of PCNA, *SRS2* cells are able to restore dsDNA and complete repair (g). In contrast to wild-type cells, in *srs2Δ* mutants, the replacement of Rad51 with RPA at the ssDNA-dsDNA junction is slow as it occurs either stochastically or relies on another, less efficient helicase (h, i). As a result, recruitment of PCNA and initiation of DNA synthesis is delayed and the ssDNA gap becomes longer (j). Once loaded, the replication machinery in *srs2Δ* mutants moves at the same rate as in wild-type cells and shortens the gap (k), albeit more DNA synthesis is now required to fill in the gap. If uninterrupted, the replication machinery will eventually catch up with the processing nucleases and complete repair, like in wild-type cells (see f and g). Re-loading of disrupted PCNA–polymerase complexes in *srs2Δ* cells depends on inefficient replacement of Rad51 with RPA at the ssDNA-dsDNA junction which leads to an increase in the ssDNA gap (l) and, depending on the balance between DNA synthesis and resection, may result in "run-away" resection and inability of *srs2Δ* to complete restoration of dsDNA (m).

During BIR, the replication machinery is involved in two distinct processes: assembly and progression of BIR forks as well as re-synthesis of resected DNA on broken chromosomes (Fig 3A). While srs2Δ cells show defects in both, we argue that the delayed progression of BIR forks has a minor effect on cell survival because by the end of the experiment the forks in srs2Δ catch up with those in SRS2 (Fig 3E, 6 h) and successfully progress almost all the way to the end of the donor chromosome (77 kb out of 94 kb to complete replication). As BIR fork migration occurs in a Rad51-free environment, the difference between SRS2 and srs2Δ could be caused by a delayed start of BIR synthesis in srs2Δ: Rad51 brought to the newly formed D-loops by invading 3′-ends might affect the recruitment of PCNA to the D-loops as it has been demonstrated in vitro (Li et al, 2013). In contrast to BIR synthesis, DNA re-synthesis was drastically reduced in srs2Δ, with the defect becoming more pronounced with the increasing distance from the break. Analysis of DNA dynamics at different positions on the resected chromosome suggests ssDNA gap "migration" away from the break site over time: while ssDNA is becoming double-stranded again closer to break, more ssDNA is produced away from the break (Fig 3C).

Break-induced replication involves resection and invasion of a one-ended DNA break. Similarly, break processing and strand-invasion operate during DSB repair by homologous recombination, but on two DNA ends. Not surprisingly, Srs2 is also required for DSB repair involving both ends (Vaze et al, 2002; Aylon et al, 2003) as re-synthesis of both processed DNA ends would be required to complete the repair.

Accumulation of ssDNA during DSB repair explains the persistence of the checkpoint activation in srs2Δ cells (Vaze et al, 2002; Yeung & Durocher, 2011). It has been suggested that cell death in srs2Δ with DSBs occurs from inability to inactivate the DNA damage checkpoint triggered by DSB processing because mec1 and mec1 srs2 cells have similar survival rates in genetic assays for DSB repair by SSA (Vaze et al, 2002). By using an inducible DSB which leads to DNA damage checkpoint activation but does not require DNA repair, we show that srs2Δ cells do not have a defect in checkpoint inactivation. Then, how do mutations in the checkpoint genes suppress the defect of srs2Δ in DSB repair? Long-range resection is known to be limited to S/G2 phase of the cell cycle but is inactive in G1 (Aylon et al, 2004; Ira et al, 2004). Lack of G2 arrest in checkpoint-deficient srs2Δ cells might terminate long-range resection by transitioning cells with unfinished repair into G1. ssDNA gaps might then be repaired in G1 or Rad51 might be removed from the DNA in an Srs2-independent manner. Alternatively, checkpoint inactivation might affect the stability of the Rad51 nucleoprotein filament in S/G2. Rad55 which is implicated in stabilization of Rad51 on DNA is phosphorylated in a Mec1-Rad53-dependent manner (Bashkirov et al, 2000) and the presence of functional Rad55/57 was reported to dictate the requirement for Srs2 in cells with DNA damage (Liu et al, 2011). Therefore, in checkpoint-deficient cells, the lack of Rad55 phosphorylation might decrease the stability of Rad51 on DNA and promote stochastic replacement of Rad51 with RPA, thereby alleviating the need for Srs2.

Srs2-dependent removal of Rad51 from ssDNA has two distinguishable functions in DNA metabolism. When Srs2 operates on Rad51 nucleoprotein filament prior to initiation of recombination

events, it acts as an anti-recombinase. However, the same enzymatic function is needed to complete recombination events when Rad51 is no longer needed on ssDNA and should be replaced by RPA in order to recruit the replication machinery and complete repair. Therefore, Srs2 can be either a pro- or an anti-recombinase. This explains the srs2Δ puzzling phenotype which includes both recombination deficiency and hyper-recombination. The anti-recombination role of Srs2 is particularly important for inhibition of recombination at replication forks and involves complex regulation of Srs2 through post-translational modifications at its C-terminus (Saponaro et al, 2010; Kolesar et al, 2012). The C-terminal part is not needed for the role of Srs2 in re-synthesis of resected DNA, and therefore, the Srs2 pro-recombination function is genetically separable from its role at replication forks. It remains uncertain if physical interactions between Srs2 and Rad51 are required for the disassembly of the Rad51 nucleofilament by Srs2. The previously reported Rad51 binding domain [a.a.875–902 (Colavito et al, 2009)] is clearly dismissible. This observation is consistent with the study by Sasanuma et al which demonstrated that Srs2 lacking the Rad51-binding domain was proficient in disruption of Rad51 filaments during meiotic recombination (Sasanuma et al, 2013). However, Srs2 contains a BRC repeat variant (BRCv, a.a.836–860). Srs2-BRCv structurally resembles the BRC repeat of the human tumour suppressor BRCA2 (Islam et al, 2012) which mediates BRCA2–RAD51 interaction, thereby promoting recruitment of RAD51 to DSBs and regulation of RAD51 recombinase activity (Jensen et al, 2010; Liu et al, 2010; Thorslund et al, 2010). Srs2-BRCv also promotes Srs2–Rad51 interaction in vitro (Islam et al, 2012). Notably, in our genetic assays, srs2 mutants lacking BRCv, Srs2(1–836), had an intermediate phenotype: they resembled wild type in SSA and BIR but were partially deficient in de novo telomere addition (Fig 8B–D). Srs2(1–836) might be partially active due to attenuated interaction with Rad51. This protein malfunction might only affect de novo telomere addition because more DNA re-synthesis might be required for completion of de novo telomere addition than BIR or SSA. Srs2(1–741) behaves like a null allele (Fig 8B–D), perhaps because the protein is no longer a functional helicase in vivo. Alternatively, the Srs2 region a.a.741–836 might be important for its interaction with Rad51 or other factors.

Recruitment of the replication machinery is evolutionarily conserved, particularly at the step of clamp loading. It has been shown that both in bacteria and humans, the clamp loaders (γ-complex and RFC) interact with single-stranded DNA binding proteins, SSB and RPA, respectively, to load clamps onto DNA (Kelman et al, 1998; Yuzhakov et al, 1999). Localization of RecA in bacteria and Rad51 in eukaryotes to SSB-/RPA-coated ssDNA leads to displacement of SSB/RPA (Kowalczykowski et al, 1987; Sugiyama & Kowalczykowski, 2002), thereby inhibiting clamp loading at repair loci until the recombination proteins are removed. Srs2 is required for Rad51 removal in yeast, UvrD is a bacterial structural homolog, and RTEL1 and RECQ5 have been suggested as functional homologs in Caenorhabditis elegans and mammals (Barber et al, 2008; Schwendener et al, 2010). FBH1 helicase is also a strong candidate for the role of Srs2 in mammals as it shares more structural similarity with Srs2 than RTEL1 and RECQ5 and has been shown to regulate RAD51 (Chu et al, 2015). Identifying human functional homolog for the role of Srs2 in restoration of resected DNA might improve not only our understanding of genome stability

mechanisms and but also human diseases stemming from defects in genome maintenance.

# Materials and Methods

### Yeast strains, oligonucleotides and plasmids

Yeast strains are described in Table EV1. Oligonucleotides are listed in Table EV2. pYT147 is described in Makovets & Blackburn (2009). To construct pYT341, the 5′-end of *srs2-K41A* was amplified by recombinant PCR: step 1a. OSM1370 + OSM1373 primers on NK1 genomic DNA; step 1b. OSM1371 + OSM1372 oligonucleotides on NK1 genomic DNA; step 2. OSM1370 + OSM1371 oligonucleotides using the mixture of step 1 fragments as a template. The final PCR product containing *srs2-K41A* was digested with *Eag*I+*Sal*I and ligated into pRS404 digested with *Eag*I+*Sal*I.

### Genetic assays

*De novo* telomere addition assay was performed as described before (Makovets & Blackburn, 2009). Briefly, cells with an inducible DSB at *MNT2* locus on chr.VIIL were patched on YPRaffinose plates and grown overnight. Cells were then resuspended in YP broth, and serial dilutions were plated on YPD and YPGalactose plates. Colonies grown on YPGalactose plates were replica plated on media without uracil. The frequency of *de novo* telomere addition was calculated as the ratio between the number of Ura⁻ colonies to the number of colonies on YPD plates (total number of cells in the experiment). The Ura⁻ colonies were also assayed by Southern blotting for the presence of a telomere at the break.

To generate a system with high frequency BIR, a 5-kb DNA sequence present on chr.IIR was placed on chr.VIIL so that chr.VIIL and chr.IIR shared extensive homology. To this end, we first placed a fragment of the *KAN-MX6* cassette linked to an HO site on chr.VIIL (*MNT2* locus) and another fragment of the same cassette at on chr.IIR (*HIS7* locus) so that when a DSB was induced on chr.VIIL, it could be repaired via BIR using the *KAN* homology on chr.IIR. BIR resulted in Kan^R colonies which contained ~100 kb of chr.IIR (*ARO4-telomere*) copied next to *MNT2*. The mutated version of chr.VIIL was then truncated by placing the HOsite-*URA3-STAR-telomere* construct next to *SPO23* (the clones were screened by PFGE to differentiate between the truncations of chr.VIIL and chr.IIR). Therefore, the two chromosomes share 6,272 bp of homology which includes incomplete *KAN-MX6* and *ARO4-SPO23* region. To calculate the efficiency of BIR, cells were patched on YPRaffinose agar and grown overnight. Yeasts were then resuspended in YP broth, and serial dilutions were plated on YPGalactose and YPD. The frequency of BIR was calculated as the ratio between the numbers of colonies on YPGalactose and YPD.

To construct a system for SSA assays, a *KAN* cassette was amplified by PCR with one of the primers containing an HO recognition sequence and the PCR product was cloned into pRS406 as an *Eag*I-*Sal*I fragment to generate a plasmid pYT381. The plasmid was linearized with *Stu*I and integrated at the *ura3-52* locus by selection on media without uracil. To assay the efficiency of SSA, cells were patched on YPRaffinose plates and grown overnight.

Cells were then resuspended in YP broth, and serial dilutions were plated on YPGalactose and YPD. Colonies grown on YPGalactose plates were replica plated on YPD+G418 plates and on media without uracil. SSA frequency was calculated as a ratio between the number of Kan^S colonies and the number of colonies on YPD plates. The frequency of SSA in the presence of HU was assayed by plating cells on YPGalactose and YPD plates containing HU at a final concentration of 5, 10 and 25 mM and calculated as described above.

### Synchronization of cell populations and DSB induction

A DSB was introduced at a genetically engineered locus by expression of the HO endonuclease placed under the galactose-inducible promoter. With the exception of the *de novo* telomere addition experiment in Fig 1D where non-synchronous populations were used, all the time-course experiments involved the following synchronization procedure. Cultures were grown at 30°C in YPRaffinose, and α-factor was added at OD$_{600}$ ~0.3 to a final concentration of 5 μg/ml for 2 h. To induce a DSB, galactose was added to the synchronized culture to a final concentration of 2% and this time point was counted as −1 h. After 1 h after the addition of galactose, cells were washed from the α-factor and released into a fresh YPGalactose medium with 15 μg/ml nocodazole to block cell division after DSB repair (time point 0 h). Cell aliquots were collected right before addition of galactose (−1 h), before the removal of α-factor (0 h) and at further interval specific to each set of experiments. For the analysis of the kinetics of SSA in the presence of HU, 2% galactose was added to the culture together with 25 mM HU. After 1 h of DSB induction, cells were released into a fresh YPGalactose medium containing 25 mM HU.

### Analysis of non-homologous DNA end cleavage during SSA by qPCR

qPCRs were performed using Brilliant II SYBR® Green QPCR Master Mix (Agilent Technologies). Each DNA sample was run in triplicates (technical repeats) in each qPCR run. A minimum of three biological repeats of each experiments (including all the strain backgrounds shown) were performed to calculate average values and standard deviations for each strain background. The kinetics of DNA cleavage at the homology to non-homology junction in SSA was quantified using two different PCRs (primer pairs OSM2233 + OSM2234 and OSM2242 + OSM2244) across the potential cleavage site. Relative amounts of DNA at the repair locus in different samples were normalized against the *ARO1* locus on chr.IVR (OSM1006 + OSM1007). The fraction of non-homologous DNA ends *remaining* was quantified relative to the −1 h time point (prior to DSB induction) using the efficiency-corrected comparative quantitation method ($\Delta\Delta C_t$) (Pfaffl, 2001). The fraction of DNA ends *cleaved* after SSA was calculated as [1—fraction of DNA ends *remaining*].

### Analysis of BIR and SSA by Southern blotting

All probes used in Southern blotting experiments were labelled using ³²P and a random prime labelling kit Prime-It II (Agilent Technologies). Phosphor-storage screens, a Typhoon Scanner and Image

Quant software (all GE Healthcare) were used for signal quantifications.

For the analysis of DSB repair via BIR, total genomic DNA was digested with *Eco*RI+*Bam*HI (NEB), resolved on a 0.7% agarose gel, transferred onto a positively charged nylon membrane (Amersham Hybond-N$^+$, GE Healthcare) and subjected to Southern blotting. To analyse the efficiency of re-synthesis of resected DNA, dsDNA fragments on chr.VII were detected using RS2.6, RS6.8 and RS15.2 hybridization probes. To analyse the efficiency of BIR, chr.IIR-specific probes BIR6, BIR36 and BIR77 were used to detect dsDNA fragments along the progression of BIR fork. A fragment on chr.V detected by the *ARS522* probe was used to normalize the amount of DNA in different samples. The efficiency of DNA re-synthesis was normalized against the −1 h time point (prior to DSB induction, 100% of DNA is dsDNA). The efficiency of BIR was calculated relative to NK3980, the control strain in which the DSB has been repaired via BIR, and therefore, it contained two copies of the *ARO4-telomere* region characteristic of chr.IIR.

For the analysis of DSB repair via SSA, total genomic DNA was digested with *Bsp*CNI+*Sma*I (fragment S1), *Bgl*II (fragment S2) or *Sal*I+*Eco*RI (fragment L), resolved on a 0.7% agarose gel, transferred onto a positively charged nylon membrane (Amersham Hybond-N$^+$, GE Healthcare) and subjected to Southern blotting. The *URA3*-specific probe was amplified using OSM2161 and OSM2162 primers and used to detect DNA fragments S1, S2 and L during repair by SSA. The *ARS1* reference probe was amplified with OSM189 and OSM190 and used to detect a reference fragment in order to normalize the amount of DNA in different samples. The efficiency of DNA repair at fragments S1, S2 and L was calculated relative to the *ura3-52* control strain NK1 (100% of dsDNA at all three fragments).

**Analysis of DNA repair via *de novo* telomere addition**

*DSB induction*
A DSB was introduced at a genetically engineered locus *MNT2:: HOsite-URA3-STAR-TEL* on the chr.VIIL by expression of the HO endonuclease placed under the galactose-inducible promoter. Cultures were grown at 30°C in YPRaffinose to the early-log phase. For the analysis of the addition of TG$_{1-3}$ repeats to a broken DNA end, a DSB was induced in the asynchronous culture by addition of galactose to a final concentration of 2%. Cultures were incubated at 30°C for 24 h. Cell aliquots were collected before DSB induction (0 h) and at certain intervals after DSB induction.

For the analysis of DNA re-synthesis after addition of telomeric repeats, cells were synchronized in G1 by addition of the α-factor at a final concentration of 5 μg/ml to a growing culture of OD$_{600}$ ~0.3 for 2 h. For DSB induction, galactose was added to a synchronized culture to a final concentration of 2%. After 1 h of DSB induction, cells were washed from α-factor, released into a fresh YPGalactose medium with 15 μg/ml nocodazole to prevent cell division after DSB repair. Cultures were incubated at 30°C for additional 6 h. Cell aliquots were collected before DSB induction (−1 h), before release into YPGalactose with nocodazole (0 h) and every hour after release into YPGalactose with nocodazole until the end of the time-course experiment.

*Detection of telomeric repeats at DSB by qPCR*
qPCRs were performed using Brilliant II SYBR® Green QPCR Master Mix (Agilent Technologies). Each DNA sample was run in triplicates (technical repeats) in each qPCR run. A minimum of three biological repeats of each experiments (including all the strain backgrounds shown) were performed to calculate average values and standard deviations for each strain background. A telomere-specific oligonucleotide OSM1487 and OSM1502 annealing 168 bp away from the HO site on chr.VIIL were used to detect *de novo* telomere addition at the induced DSB. Relative DNA amounts in different samples were normalized to the *ARO1* locus on chr.IVR (OSM1006 + OSM1007). *De novo* telomere addition was quantified relative to 0 h (prior to DSB induction) using efficiency-corrected comparative quantitation method ($\Delta\Delta C_t$) (Pfaffl, 2001).

*Quantification of ssDNA after de novo telomere addition by qPCR coupled with a restriction digest*
The protocol described by Zierhut & Diffley (2008) was adapted to calculate ssDNA/dsDNA ratios during *de novo* telomere addition. Total genomic DNA was digested with *Psi*I (NEB) which cleaves 9 bp and 51 bp away from the HO site. In the mock-digest reactions, *Psi*I was replaced with 50% (w/v) glycerol. Following digestion, samples were incubated at 65°C for 20 min to heat-inactivate the restriction enzyme and subjected to qPCR using a telomere-specific oligonucleotide OSM1487 and OSM1502 annealing 168-bp centromere-proximal to the HO site. qPCR product at the *ARO1* locus generated by using OSM1006 and OSM1007 does not have any *Psi*I sites and was used for normalization of relative DNA amounts detected in different samples.

$C_t$ values from mock-digested samples were used to quantify *de novo* telomere addition relative to the −1 h time point (prior to DSB induction) as described in the previous section. The percentage of ssDNA within each sample was quantified using the following equation:

$$ssDNA\,(undigested\,DNA),\% = \frac{100}{(1 + 2^{\Delta\Delta C_t})/2}$$

$$2^{\Delta\Delta C_t} = \frac{(1 + E(DSB))^{\Delta(DSB)}}{(1 + E(ARO1))^{\Delta C_t(ARO1)}}$$

where $2^{\Delta\Delta C_t}$: relative DNA quantity to −1 h time point; *E (DSB)*: efficiency of qPCR at the *DSB* region (OSM1487 + OSM1502); *E (ARO1)*: efficiency of qPCR at the *ARO1* locus (OSM1006 + OSM1007); $\Delta C_t = C_t$ (digested sample)—$C_t$ (undigested sample), $C_t$—threshold cycle.

To control for the efficiency of *Psi*I cleavage, *UBC5*-specific qPCRs were performed using oligonucleotides OSM2287 and OSM2288. This locus is not involved in repair (expected to contain almost 100% of dsDNA) and has one *Psi*I restriction site within the PCR template. The relative amount of DNA detected by qPCR at the *UBC5* region was normalized to the amount of DNA quantified at the *ARO1* locus, and the percentage of undigested DNA within each sample was quantified as described above. The efficiency of *Psi*I cleavage was calculated by subtracting the obtained value from 100%.

## Analysis of the DNA damage checkpoint activation after DSB induction and its effect on cell survival

Cells were grown in YPRaffinose medium at 30°C to mid-log phase. Half of the culture was subjected to DSB induction by addition of galactose to a final concentration of 2% while additional 2% raffinose was added to the remaining culture which served as negative control for DSB induction. Cell aliquots were collected 3 h after galactose addition for analysis of cell cycle distribution by flow cytometry as well as for Rad53 Western blotting. Protein extracts were run on a 6.5% SDS polyacrylamide gel and subjected to Western blotting using the anti-Rad53 primary goat antibody (Santa Cruz, 1:500) and donkey anti-goat HRP secondary antibody (Thermo Fisher Scientific, 1:6,000). To assay cell survival after DSB induction, cells were patched on YPRaffinose plates and grown overnight. Cells were then resuspended in YP broth and serial dilutions were plated on YPGalactose and YPD. The frequency of survival was calculated as a ratio between the number of colonies on YPGalactose and the number of colonies on YPD plates.

### Protein purifications

#### *Purification of PCNApka*
The plasmid (a kind gift from T. Sugiyama) expressing (His)$_6$-tagged PCNA with the protein kinase A recognition site (PCNApka) was introduced into *Escherichia coli* strain BL21(DE3). Overnight culture grown at 37°C in 2× TY medium was diluted 100-fold into fresh 2× TY medium and incubated at 37°C until OD$_{600}$~0.8. The overexpression of PCNApka protein was induced by addition of 1 mM IPTG followed by additional incubation at 37°C for 4 h. The cell pellet (9.5 g) was resuspended in CBB (50 mM Tris–HCl, pH 7.5, 10% sucrose, 2 mM EDTA) containing 600 mM KCl, 0.01% NP-40, 1 mM β-mercaptoethanol, sonicated, and centrifuged (100,000 *g*, 1 h, 4°C). Clarified supernatant was loaded onto 8-ml SP-Sepharose column with its outlet connected to a 8-ml Q-Sepharose column (GE Healthcare). Both columns were pre-equilibrated with buffer K (20 mM K$_2$HPO$_4$, 10% glycerol, 0.5 mM EDTA) containing 100 mM KCl. The Q-Sepharose column was subsequently developed with a 80-ml gradient of 100–1,000 mM KCl in buffer K. Peak fractions eluting around 400–500 mM KCl were pooled and mixed with 800 μl of His-Select Nickel Affinity Gel (Sigma) prewashed in buffer K containing 100 mM KCl for 1 h at 4°C. The beads were washed with 10 ml of 100 mM KCl in buffer K and eluted in steps with 50, 150, 300, 500 or 1,000 mM imidazole in buffer K containing 50 mM KCl. The peak fractions eluting within the range of 150–500 mM imidazole were loaded onto a 1-ml Heparin column (GE Healthcare) followed by elution using 10-ml gradient of 100–1,000 mM KCl in buffer K. The main fractions of PCNApka protein eluting around 100–400 mM KCl were pooled and applied onto a 1-ml MonoQ column (GE Healthcare) which was developed with a 9-ml gradient of 300–1,000 mM KCl. Peak fraction eluting at ~400 mM KCl was stored in small aliquots at −80°C.

#### *Purification of Srs2(1–783)*
The plasmid expressing Srs2(1–783) with (His)$_6$-affinity tag was introduced into *E. coli* strain ArcticExpress™(DE3)RIL. Protein expression was induced by 1 mM IPTG at 11°C for 24 h. Cells were lysed by sonication in CBB buffer and the lysate was clarified by

ultracentrifugation. Srs2 fragment was precipitated by adding 0.35 g/ml ammonium sulphate to the clarified supernatant. The ammonium sulphate pellet was resuspended in buffer K and incubated with 800 μl of His-Select Nickel Affinity Gel (Sigma) overnight at 4°C. The nickel beads with bound proteins were washed with 20 ml of buffer K containing 150 mM KCl and 10 mM imidazole, and eluted in steps with 50, 150, 300, and 500 mM imidazole in buffer K containing 150 mM KCl. Fractions containing Srs2 were applied onto a 1-ml Heparin column, and eluted using a 10-ml gradient of 150–1,000 mM KCl in buffer K. The peak fractions (around 400 mM KCl) were pooled, and loaded onto a 1-ml MonoS column (GE Healthcare) followed by elution using 10-ml gradient of 150–1,000 mM KCl in buffer K. Fractions containing Srs2 were pooled, concentrated to 1 μg/μl in a Vivaspin concentrator and stored at −80°C.

Rad51, RPA, RFC, PCNA, Polδ and Srs2(1–910) proteins were purified essentially as described previously (Finkelstein *et al*, 2003; Van Komen *et al*, 2006; Colavito *et al*, 2009; Sebesta *et al*, 2011).

### DNA strand extension assay

The assay was basically performed as described by Langston and O'Donnell (Langston & O'Donnell, 2008). φX174 virion circular ssDNA (0.5 nM) primed with a 70-mer (5′-CAAAACGGCAGAA GCCTGAATGAGCTTAATAGAGGCCAAAGCGGTCTGGAAACGTACG GATTGTTCAGTA-3′) was incubated with PCNA (10 nM), RFC (17.5 nM), RPA (75 nM) and Polδ (10 nM) in buffer REP (20 mM Tris–HCl pH 7.5, 1 mM DTT, 12 mM MgCl$_2$, 50 mM KCl, 0.1 μg/μl BSA, 0.09 μM dCTP, 0.09 μM dGTP, 1.25 mM ATP and an ATP-regenerating system consisting of 20 mM creatine phosphate and 20 μg/ml creatine kinase) for 5 min at 30°C. Rad51 (300 nM) was added to the indicated reactions and incubated for 5 min at 30°C followed by the addition of various amounts of Srs2. After additional incubation for 10 min at 30°C, the DNA synthesis was initiated by adding start buffer (90 μM dTTP and 0.0375 μCi [α-$^{32}$P]dATP in buffer REP). Following the incubation for 5 min at 30°C, SDS (0.5% final) and proteinase K (0.5 mg/ml) were added and mixture loaded onto an 0.8% agarose gel. After electrophoresis, the gel was dried on GRADE 3 CHR paper (Whatman), exposed to a phosphorimager screen and scanned using a Fuji FLA 9000 imager, followed by analysis with Multi Gauge software (Fuji).

### PCNA-loading assay

Phosphorylated PCNApka ($^{32}$P-labelled PCNA) was prepared essentially as described by Li and co-workers (Li *et al*, 2013). Briefly, 22 pmol of PCNApka was incubated with 1.3 pmol cAMP-dependent kinase and 2 μCi [γ-$^{32}$P]ATP in a buffer PLA (10 mM HEPES pH 7.4, 4 mM MgCl$_2$ and 2 mM DTT) at 30°C for 30 min. The $^{32}$P-labelled PCNA was stored at 4°C.

PCNA loading was analysed on a φX174 virion circular ssDNA (5,386 nt, used at 0.5 nM) primed with a 70-mer was incubated with RPA (75 nM) and/or or 2.3 μM Rad51 in buffer REP1 (20 mM Tris–HCl pH 7.5, 1 mM DTT, 12 mM MgCl$_2$, 50 mM KCl and 1 mM ATP) for 5 min at 23°C (it has been shown that Rad51 monomer binds 3 nt, meanwhile RPA heterotrimer binds 30 nt, resulting in 2.6:1 Rad51:ntDNA binding site ratio and 1:0.84 RPA:ntDNA binding site ratio, respectively). Then, $^{32}$P-PCNA (10 nM) and RFC (21 nM) were

added to the reactions followed by the incubation at 30°C for 5 min. The reactions were mixed with 0.02% glutaraldehyde and incubated for additional 10 min at 37°C, followed by addition of loading dye (60% glycerol, 10 mM Tris–HCl, pH 7.4, 60 mM EDTA and 0.025% orange G) and resolved on 0.9% agarose gel in 0.5× TBE buffer (45 mM Tris–ultrapure, 45 mM boric acid, 1 mM EDTA). After electrophoresis, the gel was analysed as in DNA strand extension assay.

**Expanded View** for this article is available online.

## Acknowledgements

We would like to thank Panayotis Filis for the pilot experiments on the project; Viktoriya Stancheva for strain construction and technical assistance; Jean-Baptiste Boulé for Pif1 protein; and Victoria Marini, Martin Pacesa and Marek Sebesta for providing various proteins used in this study. The study was funded by MRC Career Development Award to S.M. (G0900500), the Czech Science Foundation to L.K. (GACR 13-26629S and P207/12/2323), the National Program of Sustainability II (MEYS CR) to L.K. (LQ1605), and Junior researcher grant from Faculty of Medicine MU to V.A. Y.V. was a recipient of Postgraduate Fellowship from The Darwin Trust of Edinburgh.

## Author contributions

SM designed the project. YV, VA, OK, MDN, LK and SM discussed and performed the experiments and analysed the data. SM wrote the manuscript with the inputs from all other authors.

## Conflict of interest

The authors declare that they have no conflict of interest.

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
