## [Review Process File · The EMBO Journal]

Manuscript EMBO-2016-94628

Unloading of homologous recombination factors is required for restoring double-stranded DNA at damage repair loci

Yulia Vasianovich, Veronika Altmannova, Oleksii Kotenko, Matthew Newton, Lumir Krejci and Svetlana Makovets

Corresponding author: Svetlana Makovets, University of Edinburgh

Review timeline:

Submission date:	23 April 2016
Editorial Decision:	07 June 2016
Revision received:	04 October 2016
Editorial Decision:	28 October 2016
Revision received:	06 November 2016
Accepted:	08 November 2016

Editor: Hartmut Vodermaier

Transaction Report:

Additional Correspondence - Editor

30 May 2016

Thank you for submitting your manuscript EMBOJ-2016-94628, "Restoration of double-stranded DNA at repair loci requires prior unloading of homologous recombination machinery". We have now received the reports of four expert referees, which I am enclosing copied below. As you will see, the reviewers acknowledge the importance of the topic and express interest in principle in the findings and conclusions of your study. However, they also remain to be convinced that the key conclusions are supported by sufficiently strong and direct evidence to exclude alternative explanations; and they additionally raise various issues regarding descriptions and presentation in text, figures and methods. Before taking a final decision on this manuscript, I would therefore like to give you an opportunity to consider and respond to the referee reports with a brief point-by-point outline on how the major issues might be addressed/clarified; and to comment on the expected feasibility of such experiments as requested by the reviewers. These tentative responses (parts of which we may choose to share and discuss with referees) would be taken into account when making our final decision on this manuscript. I would therefore appreciate if you could send us such a response at your earliest convenience, ideally by end of this week. Should you have any further questions in this regard, please do not hesitate to let me know.

REFeree REPORTS

Referee #1:

In this manuscript Vasianovich et al. have characterised the role of the budding yeast anti-

recombinogenic helicase Srs2 on late steps of homologous recombination (HR). They have used three different *in vivo* settings to specifically monitor the HR steps that depend on DNA synthesis: 1. *de novo* telomere addition (where they monitor the synthesis of the second strand after telomerase action); 2. break-induced replication (where they monitor both the extension step and the filling of the resected regions; and 3. single-stranded annealing (where they monitor the fill-in reaction, i.e. strand extension after cleavage of the non-homologous overhangs). In all three settings they observe that deletion of SRS2 specifically affects the conversion of ssDNA to dsDNA. They hypothesize that this may be due to a failure to re-load the replication machinery in situations where the recombinogenic Rad51 filament is not removed from the ssDNA. Consistent with this, they observe an inhibition of PCNA loading by Rad51 *in vitro*. Finally, they show that deletion of EXO1, encoding a 5'-3' exonuclease, alleviates the defect of *srs2* cells, while inhibition of DNA synthesis via nucleotide depletion (hydroxyurea treatment) aggravates it. They conclude that Srs2 is needed to remove recombinogenic Rad51 from ssDNA in order to allow efficient loading of the replication machinery and restoration of dsDNA, which would in turn prevent excessive resection.

The study is interesting and novel, as it deals with a problem that is not often addressed in the recombination field, i.e. the late steps of DNA resynthesis that follow the assembly and action of the major recombination factors. In some ways, it seems logical and not all that surprising that the HR machinery needs to be removed at some point, and that this would likely be required for the late steps of DNA synthesis to ensue, but the authors do a good job of characterising these steps in molecular terms. The *in vivo* assays convincingly show the inhibition of DNA synthesis in three different, independent assays. Yet, I am not fully convinced that the conclusions from these data are as unambiguous as the authors present them. In particular, I would like to see better support for two main conclusions that are central to the authors' model:

1. The observation that Rad51 appears to directly inhibit PCNA loading is noteworthy, but I am not sure how well the data support this. The *in vitro* assays (shown in Fig. 4) were performed with a large excess of Rad51 over RPA (4-30x). Considering that the RPA-dependent stimulation of loading was needed to observe efficient loading in this assay, I am wondering whether the effect seen here is simply due to the occupation of the ssDNA by Rad51 that would then limit access of RPA and thereby prevent efficient stimulation of loading, rather than directly inhibiting the loading. I think this should be tested by using a different, neutral protein, e.g. bacterial SSB, which would likewise compete with RPA, but be unlikely to have a direct effect on the loading reaction. In addition, I would like to see a discussion of how the DNA in this assay actually looks: what is the ratio of potential binding sites to proteins (RPA and Rad51), and would one really expect mixed coverage of the DNA, or would RPA and Rad51 tend to cluster?

2. My second issue concerns the question of what actually limits Exo1 activity. The authors' central hypothesis posits that the advancing DNA synthesis limits resection, and if Rad51 is blocking the replication machinery, resection would continue to excess. I think this model should be tested by creating a situation where DNA resynthesis is blocked, but the HR machinery is removed from the DNA. If the authors' model applies, this should also lead to excessive resection. If not, one would have to conclude that removal there is another step after removal of the HR machinery that somehow quenches resection (and might then allow resynthesis).

3. It would be helpful if some of the postulated structures could be shown in a more direct way, e.g. by ChIP - if it is not possible to ChIP PCNA, maybe some other factors should be used, such as Pol delta. Also, Rad51 and RPA should be monitored in order to support the authors' model. As an alternative, fluorescence microscopy could be used to analyze the kinetics of factors assembling around the break site.

4. Fig. 1: I didn't find statistical information for the qPCR experiments.

Minor issue:

4. Fig. 3, 4: The labels are distorted.

Referee #2:

Review on Vasinovich et al.
EMBOJ-2016-94628

The paper by Vasinovich et al describes the characterization on the role of Srs2 in the DNA repair synthesis during the repair of double-strand breaks (DSBs). The authors showed that Srs2 may promote the loading of PCNA (with RFC) in the recombination intermediates in DSB repair, particularly by dismantling the Rad51 filament, which otherwise is inhibitory to the loading of the PCNA. They carried out genetic and molecular analyses using various recombination substrates as well as biochemical assays on PCNA loading. The idea is very interesting and of great interest to researchers in the field of the recombination and DNA repairs. However, the authors need more efforts to interpret their results in a fair way and should add some additional experiments to support their conclusion.

Major concerns:

1. One of the most concerns on the results is that the authors assumed extents of resection, thus formation of single-strand DNA (ssDNA) is the same among the strains used in the experiments; wild type, srs2 and rad51 mutants. The length of ssDNAs clearly affects the formation of recombination (and replication machinery) on the ssDNAs. Indeed, it is shown that the recombination defective mutants such as rad51 accumulate more resected ssDNAs in wild type. Given that the length of ssDNA directly affect the extent of DNA synthesis during the recombination, the authors essentially need to show how much ssDNAs are formed in various mutants in each assay to compensate their interpretation based on the other methods. One of the example is the result of the single-stranded annealing (SSA) assay in Figure 3D, they conclude that DNA synthesis delay in the srs2 mutant could be suppressed by the rad51 mutation. However, it is possible to interpret the results in a way that more rapid appearance of ssDNA to be annealed in the absence of the RAD51 accelerates the formation of the product.
2. In the same line, they hypothesize each event in the DSB repair is temporally separated. For example, in Figure S1A, they describe the clipping of flap strand in single-stranded annealing (SSA) pathway is AFTER the resection. However, there is no evidence on such temporal separation of these events. Even after the annealing, the resection would proceed. Thus, the authors can not say that the resection is the same in the two strains (page 4, 1st paragraph) even though the timing of cutting of non-homolog ends is almost the same. Again, it would be better to measure how much DNAs are resected in different strains seems to be critical.
3. The paper is difficult to read for general readers because of poor description on the results, particularly methods in main text, Figures and Figure legends and some errors in Figures. The authors need to be more careful in explanation on the methods. For example, in BIR assay in Figure 2, they had not mentioned HO-induced DSB is induced to initiate the event in the text and even in the Figures. Moreover, in the graphs in Figure 2B, the authors described time points studies as "Time after G1 release"??? This should be "Time after the gal induction" or "Time after DSB formation". Moreover, it would be great if the authors put the strain numbers used in each experiment in each figure legend.

Minor points:

1. Page number should be added in this manuscript, which makes hard to point out.
2. Figure 1A and 1C should be combined.
3. Figure 1F, telomere addition assay: It is important to show what happens to ssDNAs in the rad51 and rad51 srs2 mutants in this assay.
4. Figure 1F: The kinetic of dsDNA formation in the srs2 is peculiar. At 1 h time point, the srs2 mutant forms more dsDNA than the wild type, but the amount of the DNAs during further incubation is almost constant with more ssDNA formation. This may suggest two temporally separated defects in the mutant.
5. Figure 1F: X-axis should be "Time after the Gal induction".
6. Figure 2A: it would be better to describe both substrates and products with more detailed description on the site of restriction enzyme and homologous region (ARO4-SPO23) between Chromosome VIII and IIR.
7. Figure 2B and C; D and F: Blots in C should be first and followed by quantification of B.
8. Figure 2C and E; why there is no DSB bands in these blots which should be introduced by HO in this kind of BIR assay?
9. Figure 2B-E: X-axis should be "Time after the Gal induction".
10. Figure 3D: X-axis should be "Time after the Gal induction".

11. Figure 3A and C: In 3A, it would be better to show the sites of restriction enzymes and diagnostic fragments in Southern blots in Figure 3D (and S2).
12. Figure S2A: the result for rad51 and rad51 srs2 should be added for the graph in Figure 3C.
13. Figure 3F and G: This in vitro results should be described separately with results in Fig.4C-G. More importantly, as a control, it would be critical to show the Rad51 dose not inhibit DNA synthesis by Pol-delta by adding Rad51 with the polymerase.
14. Figure 3F, G, 4D, F, and H: Label of "+" and "-" are not aligned well.
15. Figure 4D, F and F: Why the PCNA complex formed in the presence of Rad51 shows slower mobility than that in the absence? This is just curiosity of mine.
16. Figure 4C-G: As a control, it would be important to show that Rad51 DOES not affect the stability of PCNA-DNA complex by adding Rad51 after the formation of the complex. In addition, given that RPA is inhibitory to Rad51-binding to ssDNA, the authors need to show increase concentrations of RPA in the system suppresses the inhibitory effect of Rad51 on this assembly reaction.

Referee #3:

Review of manuscript: "Restoration of double-stranded DNA at repair loci requires prior unloading of homologous recombination machinery" by Vasianovich et al.

In this manuscript the authors use a variety of assays (telomere addition, BIR, SSA and in vitro synthesis) to study the role of Srs2 in DSB repair. The results presented are interpreted by a model that suggests a role for Srs2 in the coordination of resection with DNA synthesis. In the absence of Srs2, Rad51 prevents PCNA loading, and DNA synthesis cannot catch up with the resection machinery leaving long patches of ssDNA. The model presented is appealing, and the results are certainly interesting. However, the data presented does not fully support the model, and additional interpretations are possible. In addition, the paper is written in a very cumbersome way that makes it difficult to follow the rationale leading to the model: Some results are not shown, others are presented succinctly as supplementary material. By and large the results are over-interpreted.

Major criticisms:

Many of the results are shown as histograms that do not show the actual data. When the data is shown, it is not always possible to see the same effects described. For example, Suppl. 2C shows no resection defects in srs2 strains compared to SRS2. This is in contrast to Fig. 1C.

Figure 1B shows a ratio between Ura- colonies and survivors on YPD. Thus, the results are affected both by the ability to add telomeres and by cell survival. It is not clear which of the two aspects are being suppressed by rad51, rad52, rad55 and rad57.

Figure 1C/D uses a PCR reaction in which one end is universal for all telomeres. This kind of reactions tend to create undesired background; the authors should show the actual results, with appropriate controls to validate that the assay works as expected. It is not clear what is the Y-axis: fragment length? Fold-elongation? This is an extremely long assay (24 hs): what happens to the cells during this long period? Are they growing? Does a differential growth ability affect the assay? And aren't est2 mutants supposed to sensesce?

Figure 1E/F again is shown as a histogram. The actual results should be shown, especially as the differences do not seem to be very large between wt and srs2 mutants.

Figure 2 shows the strongest effect of the srs2 mutant. However, there is an internal inconsistency: a gradient of effects is seen in the resection, with the defect of the srs2 mutants stronger in the 15.2, less in the 6.8 and even minor in the 2.6 kb restriction site; however, no defect whatsoever is seen in the synthesis assay. If sr2 mutants remain with ssDNA 15 kb away from the DSB 6 hs after the break, how come they have wt levels of the newly synthesized DNA at the same time? This inconsistency points to the fact that the assay is probably detecting different cells in the population, and thus making the interpretation the authors give to the results, dubious.

Figure 3D shows a dramatic effect of srs2 which is not seen in the actual blot shown in Figure Sup2. No resection problems are observed here either. Deletion of Rad51, according to this figure, restores

resection dynamics. Does it restore de novo telomere addition? Why does Rad51 deletion have no effect on SSA according to this model?

In Figure S3B, if the reason why in Δ srs2 the L fragment was restored later than wt is because of Rad51 is interfering with PCNA loading, then this delay should be abolished in a Δ rad51 strain. This has to be checked.

Figure 4B shows a delay in kinetics of srs2 strains. This is however a cell-cycle synchronized, galactose induced experiment, and thus any effect on the cell cycle or on the kinetics of DSB formation could be responsible for the observed difference. The authors should show that these parameters remain unchanged in srs2 mutants.

Although the model is in large part based on the interplay between resection and PCNA loading, much of the pertinent data is shown in supplementary figures, as an after-thought. The effect of Exo1 could be due to a large number of reasons, and no molecular data is shown to link Exo1 to the srs2 phenotypes shown. Similarly, the HU experiment could be explained in a number of ways, and it does not really support the authors' model.

Since Srs2 seems to play a role in checkpoint establishment and recovery/adaptation, many of the results presented, in particular those including Exo1 and HU, could be the result of an indirect effect of checkpoint functions.

Srs2 has been shown to recruit Exo1 (Potenski, 2014); it is not clear how this interaction fits in the model. It is also possible that this interaction is necessary to prevent unchecked Exo1 activity, and thus its deletion suppress some of srs2 phenotypes.

The Srs2 truncations include a deletion of the region required to bind Rad51 (Colavito 2012), and yet the activity is not lost. The region that interacts with PCNA is also not present. So it is not clear how PCNA loading is affected. It is also stated that: "Therefore, the role of Srs2 in DSB repair is different from its role at replication forks and does not require Srs2-PCNA interaction." The implication is that Srs2-PCNA interaction IS important during DNA replication; the reference should be given.

Referee #4:

The authors analyze the role of Srs2 in telomere addition, BIR, and SSA. They report that Srs2 is required to convert the resected DNA back to dsDNA based on molecular analysis of DNA in time courses. The results are consistent with the known role of Srs2 in removing Rad51 from ssDNA but may have uncovered the reason why srs2-deficient cells are deficient in recovery from DSB repair and adaptation to an unrepairable DSB. However, this aspect is not clearly worked out and not clearly discussed vis-à-vis the earlier work from the Haber laboratory. There are also a number of specific concerns that require clarification, additional experiments, and data analysis. The biochemical data do not really add qualitatively new information beyond what is already known from the PNAS publication by the Sugiyama lab (Li et al. 2013) except that it is shown that Srs2 can overcome the Rad51 inhibition as expected from its activity to remove Rad51 from ssDNA and the way the reactions are designed and staged in the experiments reported here.

Major comments

Figure 1 A, B: The data show that Srs2 is required for telomere addition and that this defect is suppressed by mutations in Rad51, 55, 57, and 52. It would be of significant interest to also test the Rad54 requirement to distinguish whether suppression requires absence of Rad51 filament formation or absence of recombination. Figure 1 C, D: The presentation of the data is not very clear and could be improved.

Figure 2 A-E: The data are interpreted to show a defect of srs2 mutant cells to restore the resected DSB end to dsDNA during BIR. It would help the reader to label parts B and C directly with SRS2 and srs2. If restoration from ssDNA to dsDNA would be the problem, would one not expect the signal to never drop below 50%? Instead, some bands disappear completely. It is unclear, whether

strand specific probes were used for this experiment.

Figure 3 analyzes SSA in Srs2-deficient cells. It is unclear what is plotted in B, survival or generation of physical recombinants? Please adjust labeling in F and G (also in Figure 4). Parts F and G lack quantitation. It is unclear, how the experiment can distinguish better primer usage from longer DNA synthesis without product analysis or use of end-labeled primers. Why was full-length Srs2 not used? The absence of the PCNA interaction motif likely affects the experiment.

Additional points

1) At the end of the introduction the authors discuss the anti- and pro-recombination functions of Srs2. The discussion misses important contributions by the Haber and Kupiec labs on the pro-SDSA role of Srs2 (Ira et al. 2003, Aylon et al. 2003). Moreover, it seems that the contribution by Haber (Vaze et al. 2002) is not represented adequately. The point of the Vaze et al. paper was that SSA-mediated DSB repair had no defect in srs2 mutants, but that these cells failed to recover after DSB-mediated cell cycle arrest. They also reported a defect in adaptation to an unreparable DSB. Do the authors suggest that the slow conversion to dsDNA is the root cause for the adaptation and recovery defect?

Minor comments

Page 3: line 1, Figure 1 A, B.

Page 3: spelling 'suppressed'

Page 4: s in shorter is in italics, also on this page random use of bold font.

Page 27: R of Relative is in bold

Page 29: ...using a telomere-... a and t are in bold

Page 35: meaning of letter and numbers in bold?

Additional Correspondence - Author

03 June 2016

Thank you very much for reviewing our manuscript and giving us an opportunity to respond to the Referees' comments. We are glad that all the Referees found our study novel and interesting and a lot of their comments are very useful and would help us improving the manuscript. It certainly looks like the vast majority of the comments/concerns are due to insufficient clarity in our presentation and we would address this by improving the manuscript text. If given a chance to revise we will work particularly hard on this aspect. As for now, we provide detailed explanations/clarifications for each point raised by the Referees (please see attached file).

There were some minor concerns brought up about the inability of srs2 mutants to inactivate DNA damage checkpoint and how our study relates to this srs2 phenotype (mainly Referee 4). We provide an additional set of experiments (see the text and a figure at the end of the rebuttal) addressing this particular concern.

To address other Referees' points we suggest a few additional experiments (described in more detail in the rebuttal) which we could include in a revised version:

1. ChIP on PCNA and/or Pol-delta to address replication machinery recruitment in vivo (in response to Referee 1, point 3);
2. Comparisons of DSB processing rates in WT, srs2, rad51, and rad52 srs2 and overall resection analysis (in response to Referee 1, point 2; Referee 2, points 1-2; Referee 3, one of the points);
3. Titration of RPA in vitro to outcompete Rad51 in PCNA recruitment assay (Referee 2, point 16).

If there are any further questions please don't hesitate getting back to me. Thank you once again for considering our manuscript.

Thank you for response to the referee comments on your recent submission, EMBOJ-2016-94628. I have now had a chance to consider your explanations and plans for revision, and I am pleased to see that you may be in a good position to answer the main concerns with detailed clarifications and with additional data. We shall therefore in principle be able to consider an accordingly revised version of the manuscript further for publication.

For revising the study, I agree that major emphasis should be placed on improved presentation of text and data, keeping in mind the broad readership of our journal (especially in abstract/introduction) and better explaining the experiments, their interpretation and the rationales behind them in results and discussion. Please also reorganize the figures (currently only 4 main figures and 6 supplementary figures) by capitalizing on the extended format of an EMBO Journal article, which can easily accommodate 8 or more main figures and up to 5 Expanded View figures (see below for guidelines). In particular, I feel it would be helpful to present the model in current Figure S6 as a last figure in the main manuscript.

Regarding experimental revisions, I realize that many critical points should be addressable through clarification without extra data, but it would be important to also conduct and include the additional experiments that you proposed. Furthermore, I think including the additional data on *srs2* mutants and checkpoint recovery that you obtained in the meantime would also strongly benefit the paper and help to resolve any possible discrepancies or ambiguities in light of the discussed earlier publications.

Point by point responses to referees' comments

Referee #1

(Report for Author)

In this manuscript Vasianovich et al. have characterised the role of the budding yeast anti-recombinogenic helicase Srs2 on late steps of homologous recombination (HR). They have used three different *in vivo* settings to specifically monitor the HR steps that depend on DNA synthesis: 1. *de novo* telomere addition (where they monitor the synthesis of the second strand after telomerase action); 2. break-induced replication (where they monitor both the extension step and the filling of the resected regions; and 3. single-stranded annealing (where they monitor the fill-in reaction, i.e. strand extension after cleavage of the non-homologous overhangs). In all three settings they observe that deletion of SRS2 specifically affects the conversion of ssDNA to dsDNA. They hypothesize that this may be due to a failure to re-load the replication machinery in situations where the recombinogenic Rad51 filament is not removed from the ssDNA. Consistent with this, they observe an inhibition of PCNA

loading by Rad51 *in vitro*. Finally, they show that deletion of EXO1, encoding a 5'-3' exonuclease, alleviates the defect of *srs2* cells, while inhibition of DNA synthesis via nucleotide depletion (hydroxyurea treatment) aggravates it. They conclude that Srs2 is needed to remove recombinogenic Rad51 from ssDNA in order to allow efficient loading of the replication machinery and restoration of dsDNA, which would in turn prevent excessive resection.

The study is interesting and novel, as it deals with a problem that is not often addressed in the recombination field, i.e. the late steps of DNA resynthesis that follow the assembly and action of the major recombination factors. In some ways, it seems logical and not all that surprising that the HR machinery needs to be removed at some point, and that this would likely be required for the late steps of DNA synthesis to ensue, but the authors do a good job of characterising these steps in molecular terms. The *in vivo* assays convincingly show the inhibition of DNA synthesis in three different, independent assays. Yet, I am not fully convinced that the conclusions from these data are as unambiguous as the authors present them. In particular, I would like to see better support for two main conclusions that are central to the authors' model:

1. The observation that Rad51 appears to directly inhibit PCNA loading is noteworthy, but I am not sure how well the data support this. The *in vitro* assays (shown in Fig. 4) were performed with a large excess of Rad51 over RPA (4-30x). Considering that the RPA-dependent stimulation of loading was needed to observe efficient loading in this assay, I am wondering whether the effect seen here is simply due to the occupation of the ssDNA by Rad51 that would then limit access of RPA and thereby prevent efficient stimulation of loading, rather than directly inhibiting the loading. I think this should be tested by using a different, neutral protein, e.g. bacterial SSB, which would likewise compete with RPA, but be unlikely to have a direct effect on the loading reaction. In addition, I would like to see a discussion of how the DNA in this assay actually looks: what is the ratio of potential binding sites to proteins (RPA and Rad51), and would one really expect mixed coverage of the DNA, or would RPA and Rad51 tend to cluster?

The key fact in the experiment is that ssDNA was first pre-incubated with RPA to cover ssDNA. Only after that the Rad51 protein was added to the reactions, therefore Rad51 should not limit the access of RPA in this particular setting (Rad52 would be needed in order for Rad51 to replace RPA *in vitro*). What the Reviewer is suggesting by saying “*I am wondering whether the effect seen here is simply due to the occupation of the ssDNA by Rad51 that would then limit access of RPA and thereby prevent efficient stimulation of loading, rather than directly inhibiting the loading*” is exactly what we meant by saying that the presence of Rad51 at the potential PCNA recruitment site inhibits PCNA loading. If Rad51 occupies the ssDNA binding site at the ds-ssDNA junction then RPA cannot bind there. Therefore, the RPA-dependent recruitment of PCNA to the ds-ssDNA junction does not happen until Rad51 is removed by Srs2 or leaves the binding site through other mechanisms. In short, Rad51 does not directly inhibit PCNA loading. Rather it is the absence of RPA at the junction (due to Rad51 presence) that negatively affects PCNA recruitment. We have now added an experiment where increased concentrations of RPA suppress the need for Srs2 *in vitro* (Fig 5I), further supporting the view that RPA and Rad51 are competitors for ssDNA binding. It is worth noticing though that *in vivo* Rad52 and Srs2 play opposite roles in this competition: while Rad52 promotes Rad51 replacing RPA, Srs2 indirectly promotes RPA binding by displacing Rad51.

Replacing RPA with SSB in *in vitro* PCNA recruitment assays has been done before (Figure 6 in Yuzhakov et al, 1999) where it has been shown that SSB cannot replace RPA in PCNA loading. This is consistent with the Reviewer’s suggestion and our clarifications above.

DNA substrate used in this assay is a circular single-stranded DNA (5386 nt) with annealed short complementary oligonucleotide (70 nt). The reaction mixture contained 0.5 nM DNA substrate, 75 nM RPA, or 2.3 μ M Rad51, respectively. It has been shown that Rad51 monomer binds 3 nt, meanwhile RPA heterotrimer binds 30 nt, resulting in 2.6:1 Rad51:ntDNA binding site ratio and 1:0.84 RPA::ntDNA binding site ratio, respectively. We have added this information to the manuscript (Methods, PCNA loading assay).

2. My second issue concerns the question of what actually limits Exo1 activity. The authors' central hypothesis posits that the advancing DNA synthesis limits resection, and if Rad51 is blocking the replication machinery, resection would continue to excess. I think this model should be tested by creating a situation where DNA resynthesis is blocked, but the HR machinery is removed from the DNA. If the authors' model applies, this should also lead to excessive resection. If not, one would have to conclude that removal there is another step after removal of the HR machinery that somehow quenches resection (and might then allow resynthesis).

This is another interesting comment though the proposed experiment is not straight forward as blocking repair synthesis without blocking replication overall would be hard to do. Instead of blocking re-synthesis (as reviewer suggested), we created an unrepairable DSB and monitored resection in SRS2 (where Srs2 can remove Rad51) and *srs2* over longer

stretches of DNA (see new data in Fig EV2): over the course of the experiment, resection continued as long as 20 kb away from the break in both strain backgrounds. In addition, Yeung and Durocher (2011) showed accumulation of ssDNA in srs2 cells hundreds kb away from the break which is consistent with the “run-away resection” hypothesis.

3. It would be helpful if some of the postulated structures could be shown in a more direct way, e.g. by ChIP - if it is not possible to ChIP PCNA, maybe some other factors should be used, such as Pol delta. Also, Rad51 and RPA should be monitored in order to support the authors' model. As an alternative, fluorescence microscopy could be used to analyze the kinetics of factors assembling around the break site.

Rad51 and RPA ChIP in WT and srs2 cells during SSA have been done twice (Yeung and Durocher, 2011; Esta et al., 2013). RPA does not show much difference between the two strains while Rad51 does: it accumulates over long distances in Srs2 mutants. This is consistent with the present study and our hypothesis on the role of Srs2 in DNA repair.

We agree that ChIP on PCNA or DNA polymerase would strengthen the story. We attempted to perform ChIP on both PCNA and Pol-Delta, both before the initial submission of the manuscript and again over the 4 months of revisions. Susan Gasser's lab have shown that DNA polymerases stay associated with replication forks for extended periods of time in the presence of high concentration of HU (0.2 M) and they successfully ChIPed several pol-epsilon at stalled replication forks (Cobb et al., 2003 EMBO J paper). Over the summer, we successfully reproduced their experiments on pol-epsilon, yet could not detect a good enrichment for either pol-delta or PCNA (myc tags were used in all ChIP experiments) either at stalled replication forks or at SSA loci. It could be that there is active unloading of PCNA (possibly via Elg1 as it has been suggested before) if one tries to stabilise it by any means to get ChIP experiments to work.

4. Fig. 1: I didn't find statistical information for the qPCR experiments.

The required information has been added to the Methods section (*Detection of telomeric repeats at DSB by qPCR*).

Minor issue:

4. Fig. 3, 4: The labels are distorted.

Unfortunately, PDF conversion distorted the labels and it went unnoticed. Hopefully, it does not happen again.

Referee #2

(Report for Author)

Review on Vasinovich et al.

EMBOJ-2016-94628

The paper by Vasinovich et al describes the characterization on the role of Srs2 in the DNA repair synthesis during the repair of double-strand breaks (DSBs). The authors showed that Srs2 may promote the loading of PCNA (with RFC) in the recombination intermediates in DSB repair, particularly by dismantling the Rad51 filament, which otherwise is inhibitory to the loading of the PCNA. They carried out genetic and molecular analyses using various recombination substrates as well as biochemical assays on PCNA loading. The idea is very interesting and of great interest to researchers in the field of the recombination and DNA repairs. However, the authors need more efforts to interpret their results in a fair way and should add some additional experiments to support their conclusion.

Major concerns:

1. One of the most concerns on the results is that the authors assumed extents of resection, thus formation of single-strand DNA (ssDNA) is the same among the strains used in the experiments; wild type, *srs2* and *rad51* mutants. The length of ssDNAs clearly affects the formation of recombination (and replication machinery) on the ssDNAs. Indeed, it is shown that the recombination defective mutants such as *rad51* accumulate more resected ssDNAs in wild type. Given that the length of ssDNA directly affect the extent of DNA synthesis during the recombination, the authors essentially need to show how much ssDNAs are formed in various mutants in each assay to compensate their interpretation based on the other methods. One of the example is the result of the single-stranded annealing (SSA) assay in Figure 3D, they conclude that DNA synthesis delay in the *srs2* mutant could be suppressed by the *rad51* mutation. However, it is possible to interpret the results in a way that more rapid appearance of ssDNA to be annealed in the absence of the RAD51 accelerates the formation of the product.

To address this concerns, we have added experiments on the rates of resection in WT and *srs2* over longer distances (see Fig EV2).

2. In the same line, they hypothesize each event in the DSB repair is temporally separated. For example, in Figure S1A, they describe the clipping of flap strand in single-stranded annealing (SSA) pathway is AFTER the resection. However, there is no evidence on such temporal separation of these events. Even after the annealing, the resection would proceed. Thus, the authors can not say that the resection is the same in the two strains (page 4, 1st paragraph) even though the timing of cutting of non-homolog ends is almost the same. Again, it would be better to measure how much DNAs are resected in different strains seems to be critical.

We would argue that there has to be a certain order of events in SSA. "Clipping of flap strand" is only possible after the complementary strands anneal so that the recognition substrate for Rad1/10 is formed. In turn, for the complementary strands to anneal, they both have to be single-stranded, i.e. processing has to pass through both homologous regions. Therefore, if the "clipping" step occurs without a delay in *srs2* mutants then the processing at least from the break and UP TO THE END OF HOMOLOGY has to be similar to WT cells. We agree that the processing does not stop at the end of homology and continues further and that there might be differences between WT and *srs2* further away from the break. To address this concern, we added Fig EV2, as mentioned above.

3. The paper is difficult to read for general readers because of poor description on the results, particularly methods in main text, Figures and Figure legends and some errors in Figures. The authors need to be more careful in explanation on the methods. For example, in BIR assay in Figure 2, they had not mentioned HO-induced DSB is induced to initiate the event in the text and even in the Figures. Moreover, in the graphs in Figure 2B, the authors described time points studies as "Time after G1 release"??? This should be "Time after the gal induction" or "Time after DSB formation". Moreover, it would be great if the authors put the strain numbers used in each experiment in each figure legend.

We agree with these helpful comments and did our best to improve the manuscript text by adding more explanations. All the strains used are now listed in each figure legend.

With respect to the X axis labels "Time after G1 release", they are actually accurate and explained in Methods. With the exception of the experiments presented in Figure 1D, where galactose was added to non-synchronous cells, all other time-courses were done using G1 synchronised cells. DSBs were induced in G1 arrested cells for 1 h, and then cell were released into the S-phase (with nocodazole). Because break processing and hence repair (dnta, BIR and SSA) are hardly active in G1, we assigned the point 0 h to the time of

release from G1, i.e. when the repair processes become truly activated. We stressed this point in the manuscript to make it clear.

Minor points:

1. Page number should be added in this manuscript, which makes hard to point out.

Page numbers have been added.

2. Figure 1A and 1C should be combined.

We considered this but combining these two panels makes it harder to explain things as each panel is designed so that it contains the minimum info required to explain each point.

3. Figure 1F, telomere addition assay: It is important to show what happens to ssDNAs in the rad51 and rad51 srs2 mutants in this assay.

We don't think this is necessary. The genetic data in Fig 1B (now Fig 2B) show that both rad51 and rad51srs2 cells have no problem with completing de novo telomere addition (as they form post-repair colonies in the genetic assay) and therefore we could infer that the complementary strand re-synthesis is not impaired in these strains.

4. Figure 1F: The kinetic of dsDNA formation in the srs2 is peculiar. At 1 h time point, the srs2 mutant forms more dsDNA than the wild type, but the amount of the DNAs during further incubation is almost constant with more ssDNA formation. This may suggest two temporally separated defects in the mutant.

This observation might also be explained by two classes of de novo telomeres: 1) telomeres added within the first hour of the time course to DSBs might be added prior to resection (and resection may happen later at those ends or not happen at all) and 2) de novo telomeres added to processed breaks. Unprocessed breaks have been shown (by Ira's and Haber's labs) to be very good substrates for de novo telomere addition.

5. Figure 1F: X-axis should be "Time after the Gal induction".

The X-axis is labelled accurately, see explanations above.

6. Figure 2A: it would be better to describe both substrates and products with more detailed description on the site of restriction enzyme and homologous region (ARO4-SPO23) between Chromosome VIII and IIR.

We modified Fig 2A extensively (now 3A) so that it contains a lot more details, as reviewer suggested.

7. Figure 2B and C; D and F: Blots in C should be first and followed by quantification of B.

Modified as suggested (See Fig 3).

8. Figure 2C and E; why there is no DSB bands in these blots which should be introduced by HO in this kind of BIR assay?

There are no DSB bands because we have not probed for the restriction fragment containing the HO cleavage site. We know that the HO cleavage is efficient because all the single colonies analysed post-repair have acquired a second copy of the chr. IIR 100 kb terminal fragment involved in BIR.

9. Figure 2B-E: X-axis should be "Time after the Gal induction".
10. Figure 3D: X-axis should be "Time after the Gal induction".

For both, see explanations above.

11. Figure 3A and C: In 3A, it would be better to show the sites of restriction enzymes and diagnostic fragments in Southern blots in Figure 3D (and S2).

Fig 3A (now 4A) does not have any restriction sites shown: three different restriction digests (two of them are double enzyme) were used for analysis and showing all the sites for 5 enzymes at once would clutter the schematic. Instead, we showed relevant sites and fragments before DSB induction on three derivatives of this schematic (see Fig EV3 and EV4). The positions of the sites after SSA are shown in Fig 5A and 6A to which we now added the length of the homology so that the lengths of the SSA fragments analysed could be easily calculated.

12. Figure S2A: the result for rad51 and rad51 srs2 should be added for the graph in Figure 3C.

We are not sure what is being suggested here. We assumed that the Reviewer would like us to provide rad51 and rad51srs2 blots used for quantifications in panel D. We added representative blots (see Fig EV3).

13. Figure 3F and G: This in vitro results should be described separately with results in Fig.4C-G. More importantly, as a control, it would be critical to show the Rad51 dose not inhibit DNA synthesis by Pol-delta by adding Rad51 with the polymerase.

The two in vitro assays are very different – one is an oligo extension assay and the other one is PCNA loading. We think that the logic flow is better with the present arrangement of the in vitro experiments when each of them complements the relevant in vivo data.

We don't think that addressing if Rad51 inhibits DNA synthesis in vitro is necessary as we provide more physiologically relevant in vivo evidence (Figure 5A-B) that Rad51 does not inhibit DNA synthesis per se.

14. Figure 3F, G, 4D, F, and H: Label of "+" and "-" are not aligned well.

Sorry, overlooked distortion during pdf conversion.

15. Figure 4D, F and F: Why the PCNA complex formed in the presence of Rad51 shows slower mobility than that in the absence? This is just curiosity of mine.

The more Rad51 binds to the DNA the heavier the DNA-protein complex (which includes labelled PCNA) becomes and the slower the gel mobility of the whole thing is.

16. Figure 4C-G: As a control, it would be important to show that Rad51 DOES not affect the stability of PCNA-DNA complex by adding Rad51 after the formation of the complex. In addition, given that RPA is inhibitory to Rad51-binding to ssDNA, the authors need to show increase concentrations of RPA in the system suppresses the inhibitory effect of Rad51 on this assembly reaction.

The suggested experiment on the increasing RPA concentration is presented in Fig 5I.

Referee #3

(Report for Author)

Review of manuscript: "Restoration of double-stranded DNA at repair loci requires prior unloading of homologous recombination machinery" by Vasianovich et al.

In this manuscript the authors use a variety of assays (telomere addition, BIR, SSA and in vitro synthesis) to study the role of Srs2 in DSB repair. The results presented are interpreted by a model that suggests a role for Srs2 in the coordination of resection with DNA synthesis. In the absence of Srs2, Rad51 prevents PCNA loading, and DNA synthesis cannot catch up with the resection machinery leaving long patches of ssDNA. The model presented is appealing, and the results are certainly interesting. However, the data presented does not fully support the model, and additional interpretations are possible. In addition, the paper is written in a very cumbersome way that makes it difficult to follow the rationale leading to the model: Some results are not shown, others are presented succinctly as supplementary material. By and large the results are over-interpreted.

Major criticisms:

Many of the results are shown as histograms that do not show the actual data. When the data is shown, it is not always possible to see the same effects described. For example, Suppl. 2C shows no resection defects in *srs2* strains compared to SRS2. This is in contrast to Fig. 1C.

Fig S2C (**now EV4B**) does not show resection defects, it shows *srs2* defects in the formation of the SSA repair product (see SSA/Fragment S2 arrow pointing to a band above 2 kb marker and how this band intensity becomes strong early on in WT while in *srs2*, it increases gradually over time, i.e. SSA progresses slower in the mutant). Fig 1C (**now 2C**) is a schematic for de novo telomere addition assay, not resection either. We are not sure what has been compared here.

Figure 1B shows a ratio between Ura⁻ colonies and survivors on YPD. Thus, the results are affected both by the ability to add telomeres and by cell survival. It is not clear which of the two aspects are being suppressed by *rad51*, *rad52*, *rad55* and *rad57*.

We agree that from the Fig 1B (**now 2B**) experiment it was not clear what the RAD deletions suppress as it was not clear what the *srs2* deletion affects – addition of a telomere or cell survival after the telomere addition. To further investigate this, we did the experiments shown in Fig 1C-F (**now Fig 2C-F**) which demonstrate that telomerase action at DSB is not affected by *srs2* (Fig 1C-D, **now 2C-D**) but the C-strand synthesis is (Fig. 1E-F, **now 2E-F**). This data suggest that deletions of the RAD genes suppress *srs2* by affecting processes after TG repeat addition.

Figure 1C/D uses a PCR reaction in which one end is universal for all telomeres. This kind of reactions tend to create undesired background; the authors should show the actual results, with appropriate controls to validate that the assay works as expected. It is not clear what is the Y-axis: fragment length? Fold-elongation? This is an extremely long assay (24 hs): what happens to the cells during this long period? Are they growing? Does a differential growth ability affect the assay? And aren't *est2* mutants supposed to sensesce?

Indeed, a PCR reaction with a primer annealing in multiple places could result in unwanted background. To make sure our PCR was monitoring exclusively de novo telomere addition, we uses two negative controls: one is telomerase-negative cells (*est2* deletion) which cannot add a telomere to the break due to a lack of telomerase, and the other one is Pif1-positive cells (labelled as SRS2 in Fig 1D, **now 2D**) where Pif1 efficiently inhibits telomerase at DSBs. All the qPCRs were normalised against an internal control (ARO1 locus) and then against the background (point 0). As you can see, in the two negative controls the de novo telomere-specific PCR product does not increase above the background level over the time course while in all other strains, including *pif1-m2 srs2* mutant, the levels do go up over the time suggesting that telomerase is required for the qPCR product increases in these strains.

The X-axis shows Fold increase in de novo telomere-specific PCR product relative to the background levels at point 0 h, normalised as described above. We have now clarified this in the text to avoid confusions.

We did include all the time points we had. In this assay, most cells die as they cannot repair DSBs. The cells that have repaired the break divide. We did not have to show all the time points up to 24 h – the result was clear after 6 h and even 4 h. We included all the time points that we had in order to show that de novo telomere DOES NOT increase in *est2* deletion and PIF1 controls over a long period of time. Only *rad52* deletion confers a slightly slower growth phenotype, all other strains grow at or very close to the WT rate.

Telomerase-deficient cells (*est2*) senesce after about 60 generations when telomeres become critically short. Cells used in our experiments were at 25-30th cell doubling after telomerase loss and had no growth defect at this stage of senescence.

Figure 1E/F again is shown as a histogram. The actual results should be shown, especially as the differences do not seem to be very large between wt and *srs2* mutants.

We are not sure what “actual results” should be presented but we added more info to the Methods on how qPCR experiments were done. qPCR results come as a set of Ct values deduced from amplification curves. We could provide a table of Ct values for each sample in different runs as a supplemental table. We are happy to provide any data used for the graph at 1F but it would be helpful if the Reviewer were more specific.

Figure 2 shows the strongest effect of the *srs2* mutant. However, there is an internal inconsistency: a gradient of effects is seen in the resection, with the defect of the *srs2* mutants stronger in the 15.2, less in the 6.8 and even minor in the 2.6 kb restriction site; however, no defect whatsoever is seen in the synthesis assay. If *srs2* mutants remain with ssDNA 15 kb away from the DSB 6 hs after the break, how come they have wt levels of the newly synthesized DNA at the same time? This inconsistency points to the fact that the assay is probably detecting different cells in the population, and thus making the interpretation the authors give to the results, dubious.

We need to clarify a few points here which we have explained better in the revised version of text. The graphs in Figure 2B (**now 3C**) do not monitor resection, they monitor the status of DNA on chr. VIII at 3 different positions: 2.6, 6.8, and 15.2 kb (the RS positions) from the homology with chr. IIR (grey rectangle in panel 2A). The status of DNA (single stranded or double stranded) is affected by 1) resection and 2) re-synthesis of resected DNA which both originate from the break and proceed towards the centromere (right to left in Fig 2A). Resection arrives first at RS2.6, then to RS6.8, and then to RS15.2 which is clearly observed in *srs2* mutants – at the 2 h time point the amount of dsDNA decreases much more at RS2.6 and RS6.8 than at RS15.2. At the later time-points, re-synthesis can be observed: the amount of dsDNA increases at RS2.6 in *srs2* from 4 h to 6 h, to a lesser extent this is also true for RS6.8 but not for RS15.2. We could not run the experiment for longer than 6 h as cells start escaping nocodazole arrest, but re-synthesis would have arrived at RS15.2 at later time points. Thus, in *srs2* cells there is a single stranded gap which migrates away from the break (at 2-4 h it is at RS2.6-6.8, at 4-6h it is at RS15.2) because both resection and resynthesis are moving away from the break site over time. This is consistent with the accumulation of ssDNA in *srs2* cells away from the break (Yeung and Durocher, 2011).

The BIR synthesis is shown in panel D. It is mostly independent from what is happening on the other side of homology (described above). Once a D-loop is formed, BIR proceeds towards chr.IIR telomere (left to right on the schematic in Fig 2A, **now 3A**) while re-synthesis moves in the opposite direction. From the data analysis in 2D, we conclude that the actual BIR synthesis is only mildly impaired in *srs2* mutants. Completion of BIR synthesis and inability to restore resected DNA would result in long ssDNA gaps on chr.

VII, inability to shut down the checkpoint due to this ssDNA, perhaps, inability to express some of the essential genes if they are located at the ssDNA gaps and eventual cell death. Therefore, the observed completion of BIR in *srs2* (Fig 2D, **now 3E**) does not contradict the mutants' inability to restore resected DNA (Fig. 2B, **now 3C**) as these are two separate sets of replication machinery operating on different substrates: in restoration – the substrate is ssDNA covered with RPA, recombination machinery, checkpoint proteins, etc., while in BIR – a fork progressing through dsDNA: Rad51 is not expected to localise along BIR forks and therefore Srs2 function is not very relevant there if at all.

Figure 3D shows a dramatic effect of *srs2* which is not seen in the actual blot shown in Figure Sup2. No resection problems are observed here either. Deletion of Rad51, according to this figure, restores resection dynamics. Does it restore de novo telomere addition? Why does Rad51 deletion have no effect on SSA according to this model?

Fig 3D (**now 4D**) quantifies formation of a SSA product (See SSA/Fragment L in Fig S2A), not resection. There is a clear difference between SRS2 and *srs2* in the intensity of this band (the band just above 8 kb marker): it increases gradually in *srs2* mutants while in WT it goes up rapidly and stays the same over the remaining part of the time course.

Deletion of RAD51 restores the efficiency of the fragment L formation in *srs2*, makes it even better. The gels for *rad51* mutants have been added and are presented in Fig EV3. Deletion of RAD51 also restores de novo telomere addition in *srs2* mutants (Fig. 1B, **now 2B**). *rad51* deletion has no effect on SSA in SRS2 cells (genetic assays in Fig 3B) because Rad51 is not involved in repair by SSA, only Rad52 is required (Fishman-Lobell et al., 1992; Ivanov et al., 1996).

In Figure S3B, if the reason why in Δ *srs2* the L fragment was restored later than wt is because of Rad51 is interfering with PCNA loading, then this delay should be abolished in a Δ *rad51* strain. This has to be checked.

The Reviewer's prediction is correct and this is what was tested in the experiments shown in Fig 4D (**now 5D**): the L-fragment is restored with the same dynamics in *rad51* and *rad51 srs2* cells.

Figure 4B shows a delay in kinetics of *srs2* strains. This is however a cell-cycle synchronized, galactose induced experiment, and thus any effect on the cell cycle or on the kinetics of DSB formation could be responsible for the observed difference. The authors should show that these parameters remain unchanged in *srs2* mutants.

The dynamics of DSB formation is similar in SRS2 and *srs2* cells as can be seen in Fig EV3: upon HO cleavage fragment F2a is converted into F2. We induce breaks in G1 and then WT and *srs2* cells are simultaneously released into S when similar fractions of cleaved HO-sites start being processed. Moreover, Fig. EV1 monitors SSA all the way up to the cleavage of non-homologous ends by Rad1/Rad10 and shows that up to that stage there is no difference between WT and *srs2* in the same experimental settings that we use for all SSA time courses. We also added a comparison between Srs2 and *srs2* cells in DSB resection when DSB is unreparable (Fig EV2) and show that there is no difference between the strains. We also used nocodazole to arrest repairing cells in G2. There is no reason to believe that nocodazole would work differently on SRS2 and *srs2* cells. In addition, our data from time-course experiments are consistent with the results from the genetic assays which do not involve cell cycle manipulations.

Although the model is in large part based on the interplay between resection and PCNA loading, much of the pertinent data is shown in supplementary figures, as an after-thought. The effect of Exo1 could be due to a large number of reasons, and no molecular data is shown to link Exo1 to the srs2 phenotypes shown. Similarly, the HU experiment could be explained in a number of ways, and it does not really support the authors' model.

We have reformatted the manuscript extensively and moved most of the data from supplemental to the main figures.

We respectfully disagree with the Reviewer's comments on Exo1 and particularly on HU. The effect of HU on cell physiology is well studied and it has been extensively used to slow down/pause/block replication through depletion of dNTP pools. It would be helpful if the Reviewer could provide an explanation to why the HU experiments do not support our model. Do they contradict it? What is (are) an alternative explanation to our results?

The role of Exo1 in break processing is well established and maybe on their own the Exo1 experiments don't tell a story. However, in the context of the study the Exo1 experiments as well as the HU experiments provide further insights and strengthen the evidence for our main hypothesis.

Since Srs2 seems to play a role in checkpoint establishment and recovery/adaptation, many of the results presented, in particular those including Exo1 and HU, could be the result of an indirect effect of checkpoint functions.

The concentrations of HU used (up to 25 mM) are so small that they do not cause replication forks to stall and therefore do not trigger the replication checkpoint. We provide an additional experiment demonstrating that srs2 mutants are not deficient in either checkpoint establishment or recovery from a DNA damage checkpoint (Fig 1). It is their inability to complete repair that leads to the persistence of checkpoint activation.

Srs2 has been shown to recruit Exo1 (Potenski, 2014); it is not clear how this interaction fits in the model. It is also possible that this interaction is necessary to prevent unchecked Exo1 activity, and thus its deletion suppress some of srs2 phenotypes.

The publication from the Klein lab (Potenski, 2014) on ribonucleotide removal from DNA places SRS2 and EXO1 in the same pathway, both genetically and by reporting a physical interaction between the proteins. Srs2 was found to stimulate Exo1 nucleolytic activity during RNA nucleotide removal from DNA in a pathway that is not related to DSBs or Rad51. This Exo1-stimulating activity of Srs2 is the opposite of what the Reviewer is suggesting above: Srs2 might be required "to prevent unchecked Exo1 activity", i.e. inhibit Exo1. If srs2 mutation had a major effect on resection we would have noticed it in our numerous SSA blots, as well as in the newly added experiments which directly test the effect of srs2 on DSB resection (Fig EV2).

The Srs2 truncations include a deletion of the region required to bind Rad51 (Colavito 2012), and yet the activity is not lost. The region that interacts with PCNA is also not present. So it is not clear how PCNA loading is affected.

At replication forks, PCNA recruits Srs2 by interacting with the C-terminus of Srs2, to prevent unwanted recombination, and therefore the PCNA-interacting motif at the C-terminus of Srs2 is required for the Srs2 recruitment to the forks. At the breaks, Srs2 acts UPSTREAM of PCNA: it strips Rad51 before RPA can bind the vacated ssDNA and recruit PCNA to initiate DNA synthesis. Therefore, at breaks Srs2 does not physically interact with PCNA and the C-terminus is not required for this role. We have added a Discussion section to the manuscript where this and other questions are provided with more explanations.

It is also stated that: "Therefore, the role of Srs2 in DSB repair is different from its role at replication forks and does not require Srs2-PCNA interaction." The implication is that Srs2-PCNA interaction IS important during DNA replication; the reference should be given.

Two relevant citations have been added.

Referee #4

(Report for Author)

The authors analyze the role of Srs2 in telomere addition, BIR, and SSA. They report that Srs2 is required to convert the resected DNA back to dsDNA based on molecular analysis of DNA in time courses. The results are consistent with the known role of Srs2 in removing Rad51 from ssDNA but may have uncovered the reason why srs2-deficient cells are deficient in recovery from DSB repair and adaptation to an unrepairable DSB. However, this aspect is not clearly worked out and not clearly discussed vis-à-vis the earlier work from the Haber laboratory. There are also a number of specific concerns that require clarification, additional experiments, and data analysis. The biochemical data do not really add qualitatively new information beyond what is already known from the PNAS publication by the Sugiyama lab (Li et al. 2013) except that it is shown that Srs2 can overcome the Rad51 inhibition as expected from its activity to remove Rad51 from ssDNA and the way the reactions are designed and staged in the experiments reported here.

Indeed, Sugiyama's lab studied how Rad51 affects PCNA loading (Figure 3 in Li et al., 2013). However, they used a synthetic D-loop as a substrate where RPA would not be able to bind the ss-dsDNA junction. In their schematic, they positioned RPA on ssDNA of a D-loop while Rad51 wraps the dsDNA. There is a negative effect of Rad51 on PCNA loading in their assays but it is hard to explain how Rad51 might exert it when it does not seem to compete with RPA for substrate binding (according to this model). Moreover, it is not clear if Srs2 was removing Rad51 from ssDNA or dsDNA as RPA was on ssDNA and Rad51 was shown on dsDNA. Therefore, we felt that in vitro experiments with a simpler substrate, the one closely resembling what we would predict for PCNA loading during DNA re-synthesis in vivo (at least for SSA and de novo telomere addition), would be beneficial to our study. In addition, our evidence that Srs2 can reverse the inhibitory effect of Rad51 on PCNA loading, though can be predicted from what is known, has never been reported.

Major comments

Figure 1 A, B: The data show that Srs2 is required for telomere addition and that this defect is suppressed by mutations in Rad51, 55, 57, and 52. It would be of significant interest to also test the Rad54 requirement to distinguish whether suppression requires absence of Rad51 filament formation or absence of recombination. Figure 1 C, D: The presentation of the data is not very clear and could be improved.

We agree that analysing RAD54 deletion would be very interesting, however rad54 is synthetically lethal with srs2 unless HR machinery is inactivated. Therefore, it is impossible to analyse rad54 srs2 double mutants whereas assaying rad54 srs2 rad51 (or rad52) triple would not be informative.

Figure 2 A-E: The data are interpreted to show a defect of srs2 mutant cells to restore the resected DSB end to dsDNA during BIR. It would help the reader to label parts B and C directly with SRS2 and srs2. If restoration from ssDNA to dsDNA would be the problem, would one not expect the signal to never drop below 50%? Instead, some bands disappear completely. It is unclear, whether strand specific probes were used for this experiment.

We labelled the graphs in directly with SRS2/srs2 (it is Fig 3 now).

The bands disappear almost completely (rather than up to 50%) because our method for distinguishing between ssDNA and dsDNA relies on cleavage by restriction enzymes, not on total number of DNA strands at a given locus: if DNA is single-stranded at a given locus then it can't be cleaved by a restriction enzyme. Genomic DNA has to be double stranded throughout the whole restriction fragment analysed so that the restriction enzymes used in the experiment (EcoRI and BamHI) could cut at both sites to generate a fragment of X kb which would then run on an agarose gel (a native gel in TBE) at the corresponding position and eventually hybridise to a fragment-specific probe (the probes were NOT strand-specific). If the DNA is single stranded at least at one of the restriction sites (due to resection), then such a fragment cannot be generated and the hybridisation signal in the band decreases accordingly. If more than 50% of cells have ssDNA at least at one of the restriction sites then the signal at the band corresponding to the analysed fragment drops below 50% (even below 20% in srs2 cells in 2B at 4 h). Re-synthesis of processed DNA restores dsDNA at the restriction sites and the signal "re-appears" at later time points (see Fig. 2C, RS2.6, **Fig 3 now**). If we quantified DNA by qPCR or use some sort of denaturing gels or slot-blots then the signal would have never dropped below 50% because of the remaining strand, as the Reviewer predicted.

Figure 3 analyzes SSA in Srs2-deficient cells. It is unclear what is plotted in B, survival or generation of physical recombinants? Please adjust labeling in F and G (also in Figure 4). Parts F and G lack quantitation. It is unclear, how the experiment can distinguish better primer usage from longer DNA synthesis without product analysis or use of end-labeled primers. Why was full-length Srs2 not used? The absence of the PCNA interaction motif likely affects the experiment.

It is a genetic assay in 4B. We clarified this in the text.

Part G has quantitation at the bottom of the gel, below gel lanes. The results in part F are very clear without quantitation.

Indeed, the experiments in 3F-G (now Fig 4) do not allow to distinguish between the primer usage and nucleotide incorporation. We could only state that Rad51 has an inhibitory effect on the synthesis. However, the experiments presented in Figure 5 allow to conclude that Srs2 does not affect the nucleotide incorporation in vivo (Fig 5A-B) but does affect PCNA recruitment in vitro (Fig 5D-H).

The full length recombinant Srs2 was not used because its interaction with PCNA via the C-terminus of Srs2 has a negative effect on the assay. The interaction between Srs2 and PCNA is highly regulated in vivo and perhaps in cells this negative effect is prevented through post-translational modifications of Srs2 at its C-terminus. The truncated Srs2 version has been previously used for in vitro experiments (Covalito et al., 2009).

Additional points

1) At the end of the introduction the authors discuss the anti- and pro-recombination functions of Srs2. The discussion misses important contributions by the Haber and Kupiec labs on the pro-SDSA role of Srs2 (Ira et al. 2003, Aylon et al. 2003). Moreover, it seems that the contribution by Haber (Vaze et al. 2002) is not represented adequately. The point of the Vaze et al. paper was that SSA-mediated DSB repair had no defect in srs2 mutants, but that these cells failed to recover after DSB-mediated cell cycle arrest. They also reported a defect in adaptation to an unrepairable DSB. Do the authors suggest that the slow conversion to dsDNA is the root cause for the adaptation and recovery defect?

To address the Reviewer's concerns, we brought the relevant literature to the Discussion which is now a separate, much extended section. A significant part of it is devoted to Vaze

et al., 2002 as our study provides a different view on the role of Srs2 in DSB repair. At the same time, Vaze et al. results do not contradict our main hypothesis, they just can be interpreted differently.

Yes, we do believe that accumulation of ssDNA due to slow conversion to dsDNA is the root of inability of srs2 to shut down the DNA damage checkpoint. We added an experiment (Fig 1) showing that srs2 cells have no problem recovering from a DNA damage-induced checkpoint arrest if DSBs do not have to be repaired.

Minor comments

Page 3: line 1, Figure 1 A, B.

Page 3: spelling 'suppressed'

Page 4: s in shorter is in italics, also on this page random use of bold font.

Page 27: R of Relative is in bold

Page 29: ...using a telomere-... a and t are in bold

Page 35: meaning of letter and numbers in bold?

We addressed all the minor points

2nd Editorial Decision

28 October 2016

Thank you for submitting your revised manuscript for our editorial consideration. We have now heard back from three original referees, who all consider the study significantly improved and the key concerns adequately addressed. Pending a number of remaining minor modifications, we shall therefore be happy to accept the paper for publication in The EMBO Journal.

As you will see, referee 2 still lists a few concerns, most of which can probably be addressed in writing and/or by altering presentation. Some concerns (such as point 2) would however appear to also require some additional control data, which I hope can be provided in a straightforward manner.

I am therefore returning the study to you once more for a final round of minor revision, following which we should be able to swiftly proceed with formal acceptance and publication of your manuscript.

REFeree REPORTS

Referee #1:

The authors have carried out additional experiments and have significantly reworked their manuscript in this revised version. They have addressed my major points of concern in an overall adequate manner and have provided further support for their hypotheses. Regarding my original points and the authors' reply, I would like to comment as follows:

1. The authors have now clarified that they do not imply a direct role of Rad51 on PCNA loading, but rather an indirect inhibitory role resulting from the expected and previously demonstrated competition between Rad51 and RPA. Hence, this scenario is not as novel and unexpected as it appeared to be in the original version, but the point is well taken that Srs2 - in its "classical", i.e. expected mode of action - has such effect at the late stage of DSB repair.

2. The authors' new experiment and their reference to the "run-away resection" (Yeung and Durocher 2011) are appropriate to alleviate my concern here.

3. It is unfortunate that ChIP assays were unsuccessful, as the indirect nature of most of the experiments shown in this study is still somewhat of a weak point.

4. OK

Overall, I agree with reviewer #4 that this study does not add much significant new mechanistic information about the actions of the HR machinery or Srs2 as it builds on pre-formed mechanistic concepts, but its merit lies in the careful dissection of HR stages and putting those concepts into the context of their assays. Hence, they have managed to uncover a contribution of Srs2 to the balance between resection and resynthesis, thereby explaining the defect of *srs2* mutants in "recovery" from DSB repair. I am therefore supportive of publication.

Referee #2:

Review on Vasinovich et al.
EMBOJ-2016-94628

The paper by Vasinovich et al describes the characterization on the role of budding yeast Srs2 helicase in the DNA repair synthesis during the repair of double-strand breaks (DSBs) in a way that Srs2 may promote the loading of PCNA in the recombination intermediates in DSB repair, particularly by dismantling the Rad51 filament, which otherwise is inhibitory to the loading of PCNA (by inhibiting RPA-loading). The revised version addressed most of (not all) comments by reviewers. After rewriting and revising, the paper becomes easier to read than the previous version. The content of the paper is very important to field of recombination and DNA repair and of general interest to readers in the EMBO journal. However, the authors need a bit more work to make this paper in a better shape.

Major points:

1. As described in a model in Fig. 9, the authors proposed that Srs2 can remove Rad51 to promote PCNA loading for DNA synthesis. However, at least for telomere addition and BIR, the authors need to think synthesis of RNA primers to initiate *de novo* DNA synthesis. You may need RPA for primase to synthesize RNA primers. Can the authors deny this possibility? Unfortunately, the biochemistry does not address this although they do not need to do. Moreover, results in Fig.3 support this idea. Since extension (DNA synthesis) of duplex DNA from a DSB end using invaded 3'-OH end is not severely impaired in the *srs2*. But DNA synthesis on the resected region, which seems to depend on primer RNA synthesis, is affected more.

2. As pointed by #3 reviewer, I am not convinced about the qPCR analysis for telomere addition, given the heterogeneity of telomeres length in different strains etc. The qPCR with SyberGreen is not quantitative since heterologous telomere lengths may give different fluorescent intensity (it is assumed that the length of PCR products is the same). At least the authors need to show the same telomere length in different strains. I am sorry for this additional request which was not described in my previous review. I thought the authors used Southern blotting to quantify.

3. Figure 3; Based on this result, in the *srs2* mutant, ssDNA gap is created in a region without removal of Rad51, but with extended dsDNA. Such a putative intermediate should be schematically presented to make the result easy to digest.

4. Based on results described in Fig. 7B-D, the authors should stress Rad51-BD of Srs2 is not important for the Srs2 to dismantle Rad51 filament. This is consistent with recent observation (Sasanuma et al. 2013). Rather BRCv and region of 741-836 are important for the activity. It would be great if the authors to introduce about what BRCi is in the text.

5. Discussion, page 9, second paragraph: Based on the results that the *rad51* mutant forms SS products faster than wild type (Fig.4D), the authors insist that even in wild-type cells, Rad51

inhibits DNA synthesis. This is over-interpretation. However, alternatively, it is possible that Rad51 can inhibit Rad52/RPA-mediated strand annealing or as shown, the rad51 shows faster ssDNA formation, which indirectly stimulates the SSA product formation. At least the authors carried out the exp described in Fig.EV1A using the rad51 to deny the above possibility.

Referee #3:

The authors have done a great job at answering my concerns, and most of those of the other referees. I am satisfied with this new version of the paper.

2nd Revision - authors' response

06 November 2016

Responses to Reviewer 2

Major points:

1. As described in a model in Fig. 9, the authors proposed that Srs2 can remove Rad51 to promote PCNA loading for DNA synthesis. However, at least for telomere addition and BIR, the authors need to think synthesis of RNA primers to initiate de novo DNA synthesis. You may need RPA for primase to synthesize RNA primers. Can the authors deny this possibility? Unfortunately, the biochemistry does not address this although they do not need to do. Moreover, results in Fig.3 support this idea. Since extension (DNA synthesis) of duplex DNA from a DSB end using invaded 3'-OH end is not severely impaired in the srs2. But DNA synthesis on the resected region, which seems to depend on primer RNA synthesis, is affected more.

It is a very good point - we agree with the Reviewer that primase recruitment might also be affected by Rad51 and we now brought up this point in the Discussion (p.9, Discussion, para 3 is newly added "DNA re-synthesis during BIR...").

However, we don't think that BIR DNA synthesis (Figure 3) is relevant here. We think that the srs2 mutation has little effect on BIR DNA synthesis because Rad51 does not localise ahead of the BIR fork/D-loop rather than because the primase is not involved. In fact, the primase is still required to synthesise the lagging strand during BIR but the key point is that Rad51 is not expected to localise along the path of BIR and therefore Srs2 is not needed for BIR DNA synthesis.

2. As pointed by #3 reviewer, I am not convinced about the qPCR analysis for telomere addition, given the heterogeneity of telomeres length in different strains etc. The qPCR with SyberGreen is not quantitative since heterologous telomere lengths may give different fluorescent intensity (it is assumed that the length of PCR products is the same). At least the authors need to show the same telomere length in different strains. I am sorry for this additional request which was not described in my previous review. I thought the authors used Southern blotting to quantify.

Because de novo telomere addition happens with a low frequency even in the pif1-m2 background it was not possible to use Southern blotting, qPCR was used instead as a more sensitive method (Figure 2). We would like to stress that there was no difference observed in the TG-repeat addition between WT and srs2 strains (Figure 2D). If telomerase were more processive in either of the strains, i.e. making longer de novo telomeres in one strain than the other, then we would have seen a difference between the strains progressively increasing with time. Also, there is no reason to believe that telomerase would be either more or less processive in srs2 mutants as srs2 deletion does not affect telomere length (Hedge and Klein, 2000).

The difference between WT and srs2 was observed only upon template digestion which is strictly dependent on DNA status: double-stranded vs. single-stranded. Therefore, the difference between the WT and srs2 strains is the DNA status at the de novo telomere addition site.

3. Figure 3; Based on this result, in the srs2 mutant, ssDNA gap is created in a region without removal or Rad51, but with extended dsDNA. Such a putative intermediate should be schematically presented to make the result easy to digest.

To simplify a rather busy Figure 3A we show a BIR intermediate at the stage of strand invasion, before any synthesis begins. Showing any later intermediates would complicate

the schematic further (D-loop would have to be moved forward/to the right as the synthesis progresses which will then make it more complex to show the homology region, etc.). The gap migration common to multiple repair pathways is shown in Figure 9, for both WT and srs2 mutants and this should help readers to understand the concept.

4. Based on results described in Fig. 7B-D, the authors should stress Rad51-BD of Srs2 is not important for the Srs2 to dismantle Rad51 filament. This is consistent with recent observation (Sasanuma et al. 2013). Rather BRCv and region of 741-836 are important for the activity. It would be great if the authors to introduce about what BRCi is in the text.

Added to the Discussion section according to the recommendations (p.11, last para starting from “This observation is consistent with the study by Sasanuma et al....” and to the end of the para on page 12).

5. Discussion, page 9, second paragraph: Based on the results that the rad51 mutant forms SS products faster than wild type (Fig.4D), the authors insists that even in wild-type cells, Rad51 inhibits DNA synthesis. This is over-interpretation. However, alternatively, it is possible that Rad51 can inhibit Rad52/RPA-mediated strand annealing or as shown, the rad51 shows faster ssDNA formation, which indirectly stimulates the SSA product formation. At least the authors carried out the exp described in Fig.EV1A using the rad51 to deny the above possibility.

We removed the statement challenged by the Reviewer (p.6, para 4, one sentence removed).

3rd Editorial Decision

08 November 2016

Thank you for submitting your final revised manuscript for our consideration. I am pleased to inform you that we have now accepted it for publication in The EMBO Journal.

Corresponding Author Name: Svetlana Makovets

Journal Submitted to: The EMBO Journal

Manuscript Number: EMBOJ-2016-94628